# Random Linear Projections Loss for Hyperplane-Based Optimization in Neural Networks

**Shyam Venkatasubramanian**[*1]    **Ahmed Aloui**[*1]    **Vahid Tarokh**[1]

[1] Department of Electrical and Computer Engineering, Duke University

## Abstract

Advancing loss function design is pivotal for optimizing neural network training and performance. This work introduces Random Linear Projections (RLP) loss, a novel approach that enhances training efficiency by leveraging geometric relationships within the data. Distinct from traditional loss functions that target minimizing pointwise errors, RLP loss operates by minimizing the distance between sets of hyperplanes connecting fixed-size subsets of feature-prediction pairs and feature-label pairs. Our empirical evaluations, conducted across benchmark datasets and synthetic examples, demonstrate that neural networks trained with RLP loss outperform those trained with traditional loss functions, achieving improved performance with fewer data samples, and exhibiting greater robustness to additive noise. We provide theoretical analysis supporting our empirical findings.

## 1 INTRODUCTION

Deep Neural Networks have achieved success across various applications, including computer vision [LeCun et al., 1995, Krizhevsky et al., 2012, Minaee et al., 2021], natural language processing [Hochreiter and Schmidhuber, 1997, Vaswani et al., 2017, Radford et al., 2018], generative modeling [Goodfellow et al., 2020, Kingma et al., 2019, Song et al., 2020], and reinforcement learning [Mnih et al., 2013, Van Hasselt et al., 2016, Haarnoja et al., 2018]. Foundational to these fields are the tasks of regression and classification, in which neural networks have been empirically shown to outperform conventional techniques [Reddy et al., 2012]. Training neural networks relies on the principle of empirical risk minimization (ERM) [Vapnik and Bottou, 1993], which aims to optimize the average loss on observed data to ensure model generalization. ERM relies on the development of state-of-the-art loss functions to minimize the generalization error, enabling better convergence for diverse tasks.

Among the most popular loss functions used to train neural networks are Mean Squared Error (MSE) and Cross Entropy, tailored to regression and classification tasks, respectively. MSE measures the average squared differences between the observed values (labels) and model outcomes (predictions) while Cross Entropy assesses the divergence between class labels and predicted probabilities — both MSE and Cross Entropy are measures of local pointwise deviation, as they compare individual predictions with their labels. Neural networks trained with these loss functions have achieved state-of-the-art performance across benchmark datasets for regression and classification (e.g., California Housing [Géron, 2022] and MNIST [Deng, 2012]).

Despite achieving state-of-the-art performance on benchmark datasets, neural networks trained with MSE and Cross Entropy also face significant challenges. Empirical evidence suggests these models often converge more slowly to optimal solutions, which affects training efficiency [Livni et al., 2014, Bartlett and Ben-David, 2002, Blum and Rivest, 1988]. Additionally, their performance can be limited when overparameterized [Aggarwal et al., 2018], and the presence of additive noise may result in unstable behavior and variable predictions [Feng et al., 2020]. These issues underline the limitations of neural networks optimized with loss functions akin to MSE and Cross Entropy, underscoring the necessity for more effective training methodologies.

In the deep learning literature, several methods have been proposed to address the aforementioned challenges. In particular, it is commonplace to regularize the weights of the neural network (e.g., $L_2$ regularization [Krogh and Hertz, 1991]). However, these regularization approaches usually assume the existence of a prior distribution over the model weights. Another approach is to modify the gradient descent optimization procedure itself. In particular, SGD [Rumelhart et al., 1986], SGD with Nesterov momentum [Nesterov,

---

*Equal contribution. GitHub: https://tinyurl.com/34bm8r9c.

1983], Adam [Kingma and Ba, 2014], AdamW [Loshchilov and Hutter, 2017], and Adagrad [Duchi et al., 2011] are examples of such optimizer variations. On the other hand, rather than altering the neural network training procedure, data preprocessing methods, especially data augmentation techniques, have been proven successful in computer vision, speech recognition, and natural language processing applications [Van Dyk and Meng, 2001, Chawla et al., 2002, Han et al., 2005, Jiang et al., 2020, Chen et al., 2020, Feng et al., 2021]. Among these data augmentation strategies, mixup [Zhang et al., 2017] has been proposed as a means of mitigating the vulnerabilities discussed above.

Recalling that MSE and Cross Entropy are measures of local pointwise deviation, we seek to answer a fundamental question: does the consideration of non-local properties of the training data help neural networks achieve better generalization? Firstly, as depicted in Figure 1, we note that if two functions share the same hyperplanes connecting all subsets of their feature-label pairs, then they must necessarily be equivalent. Extending this knowledge to deep learning, if the distance between sets of hyperplanes connecting fixed-size subsets (batches) of the neural network's feature-prediction pairs and feature-label pairs approaches zero, then the predicted function represented by the neural network converges to the true function (the true mapping between the features and labels). If a loss function were to incorporate this intuition, it would be able to capture non-local properties of the training data, addressing some of the limitations presented by the traditional training approach.

In this light, we introduce *Random Linear Projections (RLP) loss*: a hyperplane-based loss function that captures non-local linear properties of the training data to improve model generalization. More concretely, we consider a simple example to illustrate RLP loss. Suppose we have a training dataset consisting of $d$-dimensional features and real-valued outcomes. To train a given neural network with RLP loss, we first obtain as many fixed size ($M \geqslant d+1$) subsets of feature-label pairs as possible. Across all such subsets, we obtain a corresponding subset of feature-prediction pairs, where the predictions are the outcomes of the neural network. Subsequently, we learn the corresponding regression matrices [Van De Geer, 1987], and we minimize the distance between the hyperplanes associated with these matrices. We note that this method does not assume the true function is linear, as the large number of fixed-size subsets of feature-label pairs (random linear projections) encourages the neural network to capture potential nonlinearities.

The outline of this paper is as follows. In Section 2, we mathematically formalize RLP loss and prove relevant properties. In Section 3, we delineate the algorithm for generating fixed-sized subsets of feature-label pairs from the training data. In Section 4, we provide empirical results demonstrating that neural networks trained with RLP loss achieve superior performance when compared to MSE loss and Cross

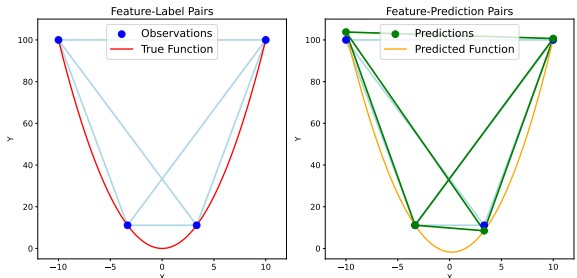

Figure 1: Comparing true and predicted functions: illustration that two functions are equivalent iff they share identical hyperplanes generated by all possible feature-label pairs.

Entropy loss. Finally, in Section 5, we summarize our work. Our contributions are summarized below:

1. We introduce *Random Linear Projections (RLP) loss*, a new loss function that leverages geometric relationships to capture non-local linear properties.

2. We prove that neural networks trained with *RLP loss* learn the optimal function when the loss is minimized, and that they converge faster than those trained with MSE loss when certain properties hold.

3. We propose an algorithmic procedure to generate fixed-size subsets of feature-label pairs that are necessary for training neural networks with *RLP loss*.

4. We demonstrate that neural networks trained with *RLP loss* achieve better performance and converge faster than those trained with MSE and Cross Entropy loss.

**Related work.** There are two primary methods for enhancing the performance of neural networks trained with MSE loss and Cross Entropy loss. On one hand, incorporating regularization during training is a prevalent approach [Wang et al., 2020, Zhang et al., 2018]. For instance, in $L_2$ regularization [Krogh and Hertz, 1991], the loss function is altered to incorporate the weighted $L_2$ norm of the weights during optimization. This discourages excessively large weights, thereby preventing overfitting. Other proposed regularization techniques include $L_1$ regularization [Tibshirani, 1996, Lv and Fan, 2009] and adaptive weight decay [Nakamura and Hong, 2019]. On the other hand, data augmentation techniques, such as mixup [Zhang et al., 2017, 2020], go beyond empirical risk minimization and have demonstrated increased robustness against noise and adversarial attacks — mixup trains a neural network on convex combinations of pairs of examples and their corresponding labels. In our study, we choose a different direction by changing the MSE loss function itself. We aim to minimize the distance between sets of hyperplanes that connect fixed-size subsets of the neural network's feature-prediction pairs and feature-label pairs. While it is conceivable to integrate both regularization and data augmentation methods into our proposed loss function, we reserve that exploration for future research.

## 2 THEORETICAL RESULTS

Let $\{(X_i, Y_i)\}_{i=1}^{M}$ denote a set of independent and identically distributed (i.i.d) random variables, where $X_i \in \mathbb{R}^d$ is the feature vector with dimension, $d$, $Y_i \in \mathbb{R}$ is the corresponding label, and $M$ is the number of considered random variables (assumed to be strictly greater than $d$). Now, let $\mathbf{X}$ denote the matrix in $\mathcal{M}_{M,d}(\mathbb{R})$ such that the $i^{\text{th}}$ row of the matrix corresponds to the vector, $X_i$. Similarly, let $\mathbf{Y}$ be the vector in $\mathbb{R}^M$ such that its $i^{\text{th}}$ element corresponds to $Y_i$.

Furthermore, we define $\mathcal{H} \subset \{h : \mathbb{R}^d \to \mathbb{R}\}$ as the class of hypothesis functions that model the relationship between $X_i$ and $Y_i$. In our empirical setting, we let $\mathcal{H}$ denote the set of neural networks that have predetermined architectures. Subsequently, we delineate $\mathbf{h} : \mathcal{M}_{M,d}(\mathbb{R}) \to \mathbb{R}^M$, where $\mathbf{X} \mapsto (h(X_1), \dots, h(X_M))^\top$ denotes the extension of the hypothesis, $h$, over the space of matrices, $\mathcal{M}_{M,d}(\mathbb{R})$.

We begin by defining the MSE loss function, the standard measure for regression tasks, and subsequently introduce our proposed Random Linear Projections (RLP) loss.

**Definition 2.1** (MSE Loss). The MSE loss function is defined as,

$$L_0(h) = \mathbb{E}\left[\|h(X) - Y\|^2\right]$$

where $(X, Y)$ and $\{(X_i, Y_i)\}_{i=1}^{M}$ are independent and identically distributed (i.i.d) random variables.

**Definition 2.2** (Random Linear Projections Loss). The RLP loss function is defined as,

$$\mathcal{L}(h) = \mathbb{E}\left[\left\|\left(\left(\mathbf{X}^\top \mathbf{X}\right)^{-1} \mathbf{X}^\top \left(\mathbf{Y} - \mathbf{h}(\mathbf{X})\right)\right)^\top X\right\|^2\right]$$

where the expectation is taken over the probability density, $p(X, X_1, Y_1, \dots X_M, Y_M)$, with $X$ being independent of and identically distributed to $\{X_i\}_{i=1}^{M}$.

The proposed definition for RLP loss is based on the observation that $\left(\mathbf{X}^\top \mathbf{X}\right)^{-1} \mathbf{X}^\top \mathbf{Y}$ and $\left(\mathbf{X}^\top \mathbf{X}\right)^{-1} \mathbf{X}^\top \mathbf{h}(\mathbf{X})$ represent the regression matrices that solve the linear problem of regressing a subset of observed outcomes and predicted outcomes, respectively, on their associated features. Consequently, RLP loss seeks to minimize the disparity between all conceivable *predicted* hyperplanes and *observed* hyperplanes. In this study, we opt to minimize the distance between these hyperplanes by evaluating the images of points drawn from the support using the random variable, $X$. This approach provides us with points from the hyperplanes, allowing us to minimize the squared distance between them. Now, we present the following proposition proving that the solution for RLP is optimal.

**Proposition 2.3.** *Let $h \in \mathcal{H}$ be a hypothesis function. We observe that $\mathcal{L}(h) \geq 0$ with the hypothesis minimizing the loss being $h(x) = \mathbb{E}[Y | X = x]$ almost surely.*

This proposition ensures that the optimal hypothesis function, $h$, aligns with the conditional expectation of $Y$ given $X = x$, almost everywhere.

Let us now consider a set of parameterized functions denoted by $\mathcal{H} = h_\theta$, where $\theta \in \Theta$. For simplicity, we represent the loss function as $\mathcal{L}(\theta)$ in place of $\mathcal{L}(h_\theta)$.

In the following proposition, we assume that the class of hypothesis functions, $\mathcal{H}$, is fully defined by a vector of parameters, $\theta \in \mathbb{R}^W$. In our empirical setting, this corresponds to the class of neural networks with predetermined architectures.

**Proposition 2.4.** *Let $L_0$ denote the MSE loss and let $\theta^*$ be the optimal parameters (i.e., $h_{\theta^*} = \mathbb{E}[Y|X]$ almost surely). We assume that both the MSE and RLP loss functions are convex. Under the following conditions:*

*(i) $\mathbb{E}[X_i X_j] = [1, \cdots, 1]^\top \mathbb{1}_{i=j}$.*

*(ii) $(\mathbf{Y} - h_\theta(\mathbf{X})) \leqslant 0$ and $\nabla_\theta h_\theta(\mathbf{X}) \leqslant 0$ (component-wise inequality).*

*(iii) For every $j, k \in \{1, 2, \dots, d\}$ and for every $l \in \{1, 2, \dots, M\}$, $\mathbb{E}[\mathbf{a}_{jk} \mathbf{a}_{lk}] \geqslant \frac{1}{d^2}$, where $(\mathbf{a}_{jk})$ and $(\mathbf{a}_{lk})$ are the components of $\mathbf{A} = \left(\mathbf{X}^\top \mathbf{X}\right)^{-1} \mathbf{X}^\top$.*

*We observe that for every step size $\epsilon \geqslant 0$ and parameter $\theta \in \mathbb{R}^W$ for which gradient descent converges,*

$$\|\theta^* - (\theta - \epsilon \nabla_\theta \mathcal{L}(\theta))\| \leqslant \|\theta^* - (\theta - \epsilon \nabla_\theta L_0(\theta))\|$$

This proposition contrasts the convergence behavior of the two loss functions, MSE and RLP, for gradient descent optimization in parameterized models. It asserts that under certain conditions — (*i*), (*ii*), and (*iii*) from Proposition 2.4 — updates based on the gradient of the RLP loss function bring the parameters closer to the optimal solution than those based on the gradient of the MSE loss function.

## 3 ALGORITHM

In this section, we detail our methodology for training neural networks using the Random Linear Projections (RLP) loss. Our approach comprises two main steps. First, we employ the *balanced batch generation* strategy to sample unique batches from the training dataset. Subsequently, we utilize these batches to train a neural network model using gradient descent and our proposed RLP loss.

Let $J = \{(x_i, y_i)\}_{i=1}^{N}$ denote the observed training dataset, where $x_i \in \mathbb{R}^d$ and $y_i \in \mathbb{R}$. Let $M \ll N$ be the number of training examples used to identify the regression matrices of the different hyperplanes, where $M$ is denoted as the batch size. Let $P = \frac{N!}{(M+1)!(N-M-1)!}$. The RLP loss is computed by examining all possible combinations of size $M + 1$ from the training data. For each combination, regression matrices

are constructed using the first $M$ components. Subsequently, the dot product is calculated between this regression matrix and the $(M+1)^{\text{th}}$ component. Hence the proposed empirical RLP loss function can be defined as follows:

$$L(\theta) = \frac{1}{P} \sum_{j=1}^{P} \left( \left( \left( \mathbf{x}_j^\top \mathbf{x}_j \right)^{-1} \mathbf{x}_j^\top \left( \mathbf{y}_j - \mathbf{h}_\theta(\mathbf{x}_j) \right) \right)^\top x_j \right)^2$$

Above, $\mathbf{x}_j = (x_{j_1}, \ldots, x_{j_M})^\top$ is the matrix in $\mathcal{M}_{M,d}(\mathbb{R})$, whose rows correspond to $M$ different $x_{j_k}$ from the set of training data feature vectors, $\mathbf{y}_j = (y_{j_1}, \ldots, y_{j_M})^\top$ denotes the corresponding labels, and $x_j$ denotes an observed feature vector distinct from all rows comprising matrix $\mathbf{x}_j$. It is important to note that by invoking the law of large numbers, the empirical RLP will converge in probability to the RLP loss (Definition 2.2). Given that the number of permutations can be exceedingly large, our approach for training the regression neural network with the RLP loss involves randomly sampling $K$ batches from the $P$ possible batches of size $M$ that comprise the training dataset, $J$.

### 3.1 BALANCED BATCH GENERATION

The objective of *balanced batch generation* is to produce batches from the training dataset such that each example appears in at least one batch, where no two batches are identical. Let $J$ denote the training dataset, with corresponding labels, $M$ be the size of each batch, and $K$ be the total number of batches we intend to generate. To construct balanced batches, $\mathcal{B}$, from $J$, our methodology involves a continuous sampling process, ensuring each data point is incorporated in at least one batch. To maintain the uniqueness of batches and avoid repetitions, we employ a tracking set, $\mathcal{I}$.

---

**Algorithm 1:** Balanced Batch Generator

**Input:** $J$ (Training dataset), $M$ (Batch size), $K$ (Number of batches to generate)

**Output:** $\mathcal{B}$ (Set of generated batches)

1   $\mathcal{I} \leftarrow \{0, 1, \ldots, |J| - 1\}$ (Initialize set of all indices)
2   $\mathcal{B} \leftarrow \emptyset$ (Initialize set of generated batches)
3   **while** $|\mathcal{B}| < K$ **do**
4      Randomly shuffle $\mathcal{I}$ to obtain $\mathcal{I}_{\text{shuffled}}$
5      **for** $i = 0, M, 2M, \ldots, |J| - M$ **do**
6          $b \leftarrow \{J[\mathcal{I}_{\text{shuffled}}[i : i + M]]\}$
7          **if** $b \notin \mathcal{B}$ **then**
8              $\mathcal{B} \leftarrow \mathcal{B} \cup \{b\}$
9          **if** $|\mathcal{B}| \geqslant K$ **then**
10            **break**

11   **return** $\mathcal{B}$

---

The main loop facilitates the consistent sampling of unique batches until we accumulate a total of $K$ batches. Within

this loop, we first generate a full sequence of dataset indices, followed by a shuffle operation to ensure randomness. Iteratively, we then allocate train examples to batches in strides of size $M$. As each batch is formed, we check for its existence within our $\mathcal{I}$ set to uphold the uniqueness principle. This operation continues until we have attained our target number of unique batches, $K$.

Per Algorithm 1, we observe that each training example in $J$ appears in at least one batch and that no two batches in $\mathcal{B}$ are identical. Subsequently, during each training epoch, we iterate over the $K$ randomly sampled batches and employ the Random Linear Projections loss. Subsequently, the algorithm for training a neural network using gradient descent with the RLP loss is provided in Algorithm 2.

---

**Algorithm 2:** Neural Network Training With RLP Loss

**Input:** $J$ (Training dataset), $\theta$ (Initial NN parameters), $\alpha$ (Learning rate), $M$ (Batch size), $K$ (Number of batches to generate), $E$ (Number of epochs)

**Output:** $\theta$ (Trained NN parameters)

1   $\mathcal{B} \leftarrow \texttt{Balanced\_Batch\_Generator}(J, M, K)$
2   **for** *epoch* $= 1, 2, \ldots, E$ **do**
3      **for** $j = 1, 2, \ldots, K$ **do**
4          $\mathbf{x}_j \leftarrow$ Matrix of features from batch $\mathcal{B}[j]$
5          $\mathbf{y}_j \leftarrow$ Vector of labels from batch $\mathcal{B}[j]$
6          $M_y \leftarrow \left( \mathbf{x}_j^\top \mathbf{x}_j \right)^{-1} \mathbf{x}_j^\top \mathbf{y}_j$
7          $M_h \leftarrow \left( \mathbf{x}_j^\top \mathbf{x}_j \right)^{-1} \mathbf{x}_j^\top \mathbf{h}_\theta(\mathbf{x}_j)$
8          Randomly sample $x_j$ (feature vector) from $J$
9          $l_j(\theta) \leftarrow \left( (M_y - M_h)^\top x_j \right)^2$
10      $\mathcal{L}(\theta) \leftarrow \frac{1}{K} \sum_{j=1}^{K} l_j(\theta)$
11      $\theta \leftarrow \theta - \alpha \nabla_\theta \mathcal{L}(\theta)$
12   **return** $\theta$

---

The above algorithm, Algorithm 2, provides a systematic procedure for training a neural network with RLP loss. By iterating through each epoch, and for each batch within this epoch, we compute the observed regression matrix, calculate the RLP loss, and then update the model parameters using gradient descent. This iterative process continues for a predefined number of epochs, ensuring that the model converges to a solution that minimizes the RLP loss.

## 4 EMPIRICAL RESULTS

In this section, we present our empirical results for regression, image reconstruction, and classification tasks, using a variety of synthetic and benchmark datasets. We first present the regression results on two benchmark datasets (California Housing [Géron, 2022] and Wine Quality [Cortez et al., 2009]), as well as two synthetic datasets: one *Linear* dataset where the true function is a linear combination of the fea-

tures in the dataset, and one *Nonlinear* dataset, where the true function combines polynomial terms with trigonometric functions of the features in the dataset. For the image reconstruction tasks, we utilize two different datasets: MNIST [Deng, 2012] and CIFAR10 [Krizhevsky et al., 2009]. We also present the classification results on MNIST (for classification results on the Moons dataset, see Section C.1 of the Appendix). A comprehensive description of these datasets is provided in Section B of the Appendix.

For the evaluations that follow, our default setup utilizes $|J| = 0.5|\mathcal{X}|$ training and $|G| = |\mathcal{X}| - |J|$ test examples, where $|\mathcal{X}|$ signifies the size of each dataset — we do not consider any distribution shift or additive noise. To ensure a fair comparison, we also use the same learning rate for each loss and network architecture across all experiments on a given dataset. Deviations from this configuration are explicitly mentioned in the subsequent analysis. We first present the performance results when the neural network is trained with RLP loss, MSE loss, and MSE loss with $L_2$ regularization for regression and reconstruction tasks, and with RLP loss and Cross Entropy loss for classification tasks. For the RLP loss case, the neural network is trained using $K = 1000$ batches (see Algorithm 1). Moreover, we present ablation studies on the impact of three different factors:

(1) The number of training examples $|J| \in \{50, 100\}$.

(2) The distribution shift bias $\gamma \in \{0.1, 0.2, ..., 0.9\}$.

(3) The noise scaling factor $\beta \in \{0.1, 0.2, ..., 0.9\}$ for the additive standard normal noise.

In **(1)**, the neural network is trained using $K = 100$ batches for the RLP case, and in **(2)** and **(3)**, the neural network is trained using $K = 1000$ batches for the RLP case, produced via Algorithm 1. Our empirical findings demonstrate that the proposed loss helps mitigate the vulnerability of neural networks to these issues.

## 4.1 PERFORMANCE ANALYSIS

This first evaluation provides an in-depth assessment of our methods across various benchmark and synthetic datasets, illuminating the efficacy of RLP loss compared to MSE loss, its variant with $L_2$ regularization, and Cross Entropy loss, when there are no ablations introduced within the data.

**Regression Task Results.** For the California Housing dataset, a benchmarking dataset for regression tasks, we observe several differences in performance. In particular, when we leverage the Adam optimizer [Kingma and Ba, 2014], the regression neural network trained with RLP loss demonstrates enhanced efficacy. This contrasts with the case where the regression neural network is trained with MSE loss and its $L_2$ regularized counterpart. Notably, RLP loss not only exhibits superior performance, since the test error is lower when measured using MSE or RLP, but also

demonstrates resilience against overfitting. We observe that after 100 training epochs, MSE loss and its $L_2$ regularized counterpart begin to overfit the training data, resulting in diminished generalization, whereas RLP loss continues to minimize the test error. We further observe that the standard deviation of the test error (compiled after 500 training epochs) is demonstrably lower when the regression neural network is trained with RLP loss as opposed to MSE loss or MSE loss with $L_2$ regularization.

Subsequently, for the Wine Quality dataset, which has features derived from physicochemical tests assessing wine constituents and their influence on quality, we discern several performance differences. When using the Adam optimizer, the regression neural network trained with RLP loss outperforms those trained with MSE loss and $L_2$ regularized MSE loss. RLP loss not only showcases improved performance metrics — evidenced by a reduced test error when assessed by either MSE or RLP — but also demonstrates more rapid convergence. In particular, within just 20 training epochs, we observe a test MSE of 0.6 in the RLP loss case. In contrast, both the MSE loss and MSE loss + $L_2$ regularization cases only achieve this test MSE after 200 epochs. Moreover, we find that the standard deviation of the test error, gathered after 200 training epochs, is lower when the regression neural network is trained using RLP loss as opposed to MSE loss or its $L_2$ regularized variant.

Delving into the synthetic datasets, we consider the aforementioned two scenarios: a Linear dataset, where the true function is a linear combination of its features, and a Nonlinear dataset, where the true function blends polynomial and trigonometric functions of its features. Using the Adam optimizer, the regression neural network trained with RLP loss exhibits a notably improved performance trajectory in both synthetic scenarios, outpacing networks trained with MSE loss and $L_2$ regularized MSE loss. In the context of the Linear dataset, the efficacy of RLP loss is particularly pronounced, as it converges to a test MSE below $3 \times 10^{-6}$ within 200 training epochs. In contrast, both MSE loss and its $L_2$ regularized version yield a test MSE above 0.2 at the same epoch count. As it pertains to the Nonlinear dataset, RLP loss similarly yields a lower test error in comparison to MSE loss and its $L_2$ regularized counterpart. Cumulatively, these results demonstrate that RLP loss yields improved performance over MSE loss and MSE loss with $L_2$ regularization, even when the true function has nonlinearities.

**Image Reconstruction Task Results.** With MNIST, a benchmark dataset in image reconstruction tasks, our findings are in line with previous observations. Leveraging the SGD optimizer with Nesterov momentum [Nesterov, 1983], we observe that the autoencoder trained using RLP loss yields a test MSE of 0.018 after 100 training epochs, whereas the autoencoder trained using MSE loss or $L_2$ regularized MSE loss yields a test MSE above 0.04 after 100

Table 1: Test performance across different datasets for $|J| = 0.5|\mathcal{X}|$ training examples and $|\mathcal{X}| - |J|$ test examples.

| Dataset | MSE Perf. (MSE) | MSE Perf. (RLP) | MSE with $L_2$ Perf. (MSE) | MSE with $L_2$ Perf. (RLP) | RLP Perf. (MSE) | RLP Perf. (RLP) |
|---|---|---|---|---|---|---|
| California Housing | $0.915_{\pm 0.997}$ | $0.101_{\pm 0.127}$ | $0.961_{\pm 1.151}$ | $0.106_{\pm 0.102}$ | $0.575_{\pm 0.314}$ | $0.016_{\pm 0.012}$ |
| Wine Quality | $0.542_{\pm 0.014}$ | $0.194_{\pm 0.386}$ | $0.546_{\pm 0.015}$ | $0.169_{\pm 0.089}$ | $0.532_{\pm 0.011}$ | $0.031_{\pm 0.015}$ |
| Linear | $0.227_{\pm 0.104}$ | $0.111_{\pm 0.059}$ | $0.209_{\pm 0.086}$ | $0.087_{\pm 0.065}$ | $2.6e\text{-}6_{\pm 1.7e\text{-}6}$ | $5.2e\text{-}8_{\pm 9.2e\text{-}8}$ |
| Nonlinear | $0.075_{\pm 0.009}$ | $0.008_{\pm 0.004}$ | $0.073_{\pm 0.006}$ | $0.008_{\pm 0.005}$ | $0.033_{\pm 0.012}$ | $0.002_{\pm 0.001}$ |
| MNIST | $0.042_{\pm 0.001}$ | $0.047_{\pm 0.001}$ | $0.052_{\pm 0.011}$ | $0.049_{\pm 0.001}$ | $0.018_{\pm 0.002}$ | $4.7e\text{-}3_{\pm 5.0e\text{-}4}$ |
| CIFAR-10 | $6.0e\text{-}4_{\pm 1.0e\text{-}4}$ | $6.1e\text{-}4_{\pm 9.8e\text{-}6}$ | $1.8e\text{-}3_{\pm 1.0e\text{-}4}$ | $1.8e\text{-}3_{\pm 7.1e\text{-}6}$ | $2.7e\text{-}5_{\pm 1.0e\text{-}6}$ | $2.7e\text{-}5_{\pm 1.0e\text{-}6}$ |

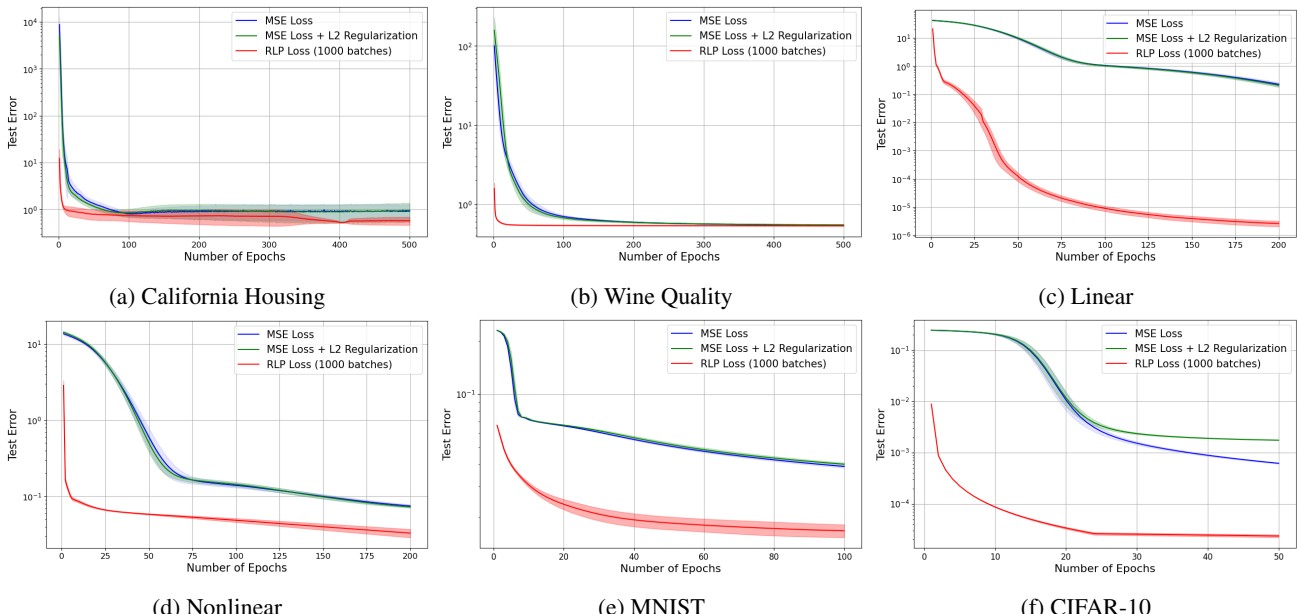

(a) California Housing     (b) Wine Quality     (c) Linear

(d) Nonlinear     (e) MNIST     (f) CIFAR-10

Figure 2: Test performance comparison across six datasets (California Housing, Wine Quality, Linear, Nonlinear, MNIST, and CIFAR-10) using three different loss functions: Mean Squared Error (MSE), MSE with $L_2$ regularization (MSE + $L_2$), and RLP. The x-axis represents training epochs, while the y-axis indicates the test MSE.

training epochs. When the test error is instead measured using RLP, the gains provided by RLP loss over MSE and MSE + $L_2$ regularization become even more apparent. As in before, we also observe that the standard deviation of the test error, gathered after 100 training epochs, is lower when the autoencoder is trained using RLP loss instead of MSE loss or its $L_2$ regularized variant.

Our experiments on CIFAR-10 corroborate our earlier findings from the MNIST experiments. Utilizing the SGD optimizer with Nesterov momentum, we observe a test MSE of $2.7 \times 10^{-5}$ when the autoencoder is trained with RLP loss for 50 epochs. In contrast, we observe a test MSE exceeding $5.0 \times 10^{-4}$ when the autoencoder is trained with MSE loss or $L_2$ regularized MSE loss for 50 epochs. We also observe a reduction in the standard deviation of the test error after 50 epochs when the autoencoder is trained using RLP loss versus MSE loss and MSE loss with $L_2$ regularization.

**Classification Task Results.** Per Section C.1 of the Appendix, RLP loss can also be applied to classification tasks. We consider the MNIST dataset for our experiments. Using the AdamW optimizer Loshchilov and Hutter [2017], we observe that the convolutional neural network (CNN) converges to a test accuracy of 96% after 10 epochs using RLP loss. In contrast, we observe a test accuracy of 86% when the CNN is trained with Cross Entropy loss after 10 epochs. This evaluation demonstrates that the faster convergence yielded by RLP loss is preserved in classification scenarios.

### 4.2 ABLATION STUDIES

This next evaluation delves into the performance dynamics of our methods under ablated data scenarios, highlighting the resilience of RLP loss relative to MSE loss and MSE loss with $L_2$ regularization in the presence of data perturbations.

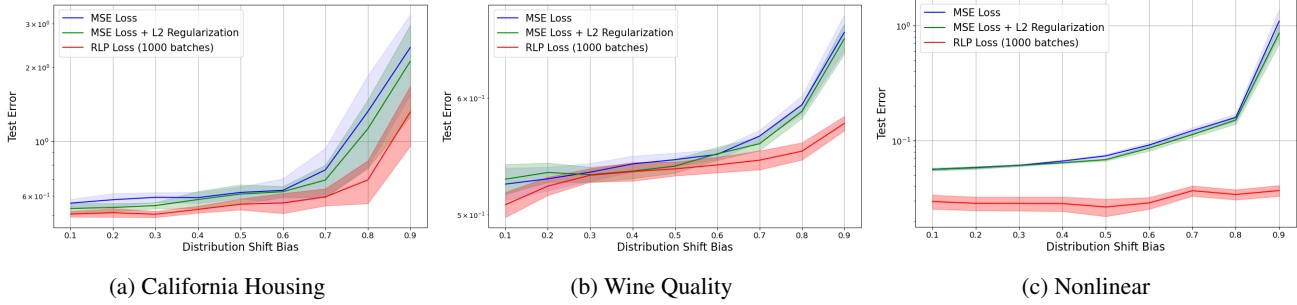

| (a) California Housing | (b) Wine Quality | (c) Nonlinear |

Figure 3: Distribution shift test performance comparison across three datasets (California Housing, Wine Quality, and Nonlinear) using three different loss functions: Mean Squared Error (MSE), MSE with $L_2$ regularization (MSE + $L_2$), and RLP. The x-axis is the degree of bias, $\gamma$, between the test data and the train data, while the y-axis indicates the test MSE.

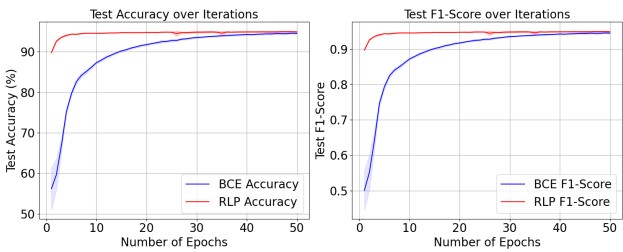

Figure 4: Test performance comparison on MNIST using Cross Entropy loss and RLP loss. The x-axis represents training epochs, while the y-axis indicates the classification accuracy (left) and F1 score (right).

Table 2: Test Performance for $|J| = 50$ training examples.

| Dataset | MSE | MSE+$L_2$ | RLP |
|---|---|---|---|
| Cali. Housing | $4.09_{\pm 3.00}$ | $4.42_{\pm 3.44}$ | $3.04_{\pm 1.87}$ |
| Wine Quality | $1.16_{\pm 0.26}$ | $1.31_{\pm 0.47}$ | $1.15_{\pm 0.14}$ |
| Linear | $0.86_{\pm 0.19}$ | $0.84_{\pm 0.20}$ | $5.0\text{e-}4_{\pm 7.0\text{e-}4}$ |
| Nonlinear | $0.13_{\pm 0.02}$ | $0.13_{\pm 0.03}$ | $0.09_{\pm 0.03}$ |
| MNIST | $0.23_{\pm 0.01}$ | $0.23_{\pm 0.01}$ | $0.05_{\pm 0.01}$ |
| CIFAR-10 | $0.25_{\pm 0.01}$ | $0.25_{\pm 0.01}$ | $5.6\text{e-}4_{\pm 1.2\text{e-}5}$ |

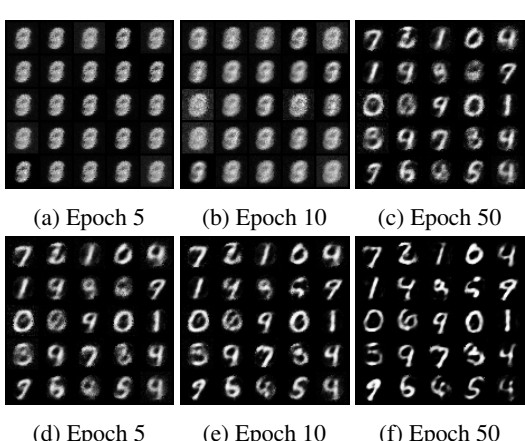

| (a) Epoch 5 | (b) Epoch 10 | (c) Epoch 50 |

| (d) Epoch 5 | (e) Epoch 10 | (f) Epoch 50 |

Figure 5: Comparison of reconstructed images for an autoencoder trained with MSE loss (top row) and RLP loss (bottom row) at different epochs. The model trained with RLP loss learns faster and better with limited data ($|J| = 50$).

**Number of Training Examples.** In the ablation study with a constraint of $|J| = 50$ training examples, our findings across the six datasets underline the robustness and efficacy of the RLP loss. For the California Housing and Wine Quality regression benchmark datasets, the RLP loss-trained models consistently outperform their MSE loss and $L_2$ regularized MSE loss-trained counterparts in both con-

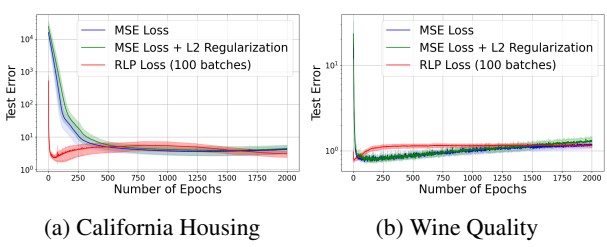

| (a) California Housing | (b) Wine Quality |

Figure 6: Limited training data ($|J| = 50$) test performance comparison across two datasets (California Housing and Wine Quality) using three different loss functions: Mean Squared Error (MSE), MSE with $L_2$ regularization (MSE + $L_2$), and RLP. The x-axis represents training epochs, while the y-axis indicates the test MSE.

vergence rate and test error, despite the limited data. This trend persists in the Linear and Nonlinear synthetic regression datasets, with RLP loss-trained neural networks achieving rapid convergence and low error. Similarly, for the image reconstruction tasks on MNIST and CIFAR-10, RLP loss-trained models achieve faster convergence. Figure 5 in particular shows that after 5, 10, and 50 training epochs, the MNIST images reconstructed by the RLP loss-trained autoencoder are more accurate and clearer than those generated by the MSE loss-trained autoencoder. Results for the case of $|J| = 100$ training data points can be found in Section C.2 of the Appendix.

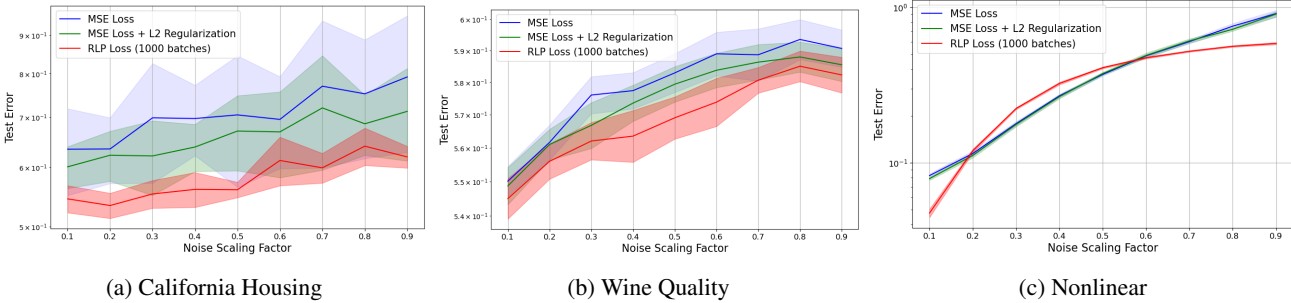

(a) California Housing       (b) Wine Quality       (c) Nonlinear

Figure 7: Noise robustness test performance comparison across three datasets (California Housing, Wine Quality, and Nonlinear) using three different loss functions: Mean Squared Error (MSE), MSE with $L_2$ regularization (MSE + $L_2$), and RLP. The x-axis is the scaling factor, $\beta$, for the additive standard normal noise, while the y-axis indicates the test MSE.

**Distribution Shift Bias.** In this ablation study, we consider the case of a distribution shift between the train and test data, characterized by a bias parameter, $\gamma$. Given a dataset $\mathcal{X}$ consisting of $d$-dimensional feature vectors, $x_i$, let $\boldsymbol{\mu}$ be the mean vector of $\mathcal{X}$ and $\boldsymbol{\sigma}$ be the standard deviation vector. Regarding preliminaries, we introduce a notation for element-wise comparison of vectors: for two vectors $\mathbf{a}, \mathbf{b} \in \mathbb{R}^d$, we write $\mathbf{a} \prec \mathbf{b}$ to denote that $a_j < b_j$ for all $j \in \{1, 2, \ldots, d\}$. Using this notation, we define the region of interest (ROI) in the feature space via two conditions that must hold simultaneously: $x_i - \boldsymbol{\mu} \prec \boldsymbol{\epsilon}$ and $\boldsymbol{\mu} - x_i \prec \boldsymbol{\epsilon}$, where $\boldsymbol{\epsilon} = 0.5 \times \boldsymbol{\sigma}$. Per these definitions:

**(1)** For examples within the ROI (close to the mean):

$$\mathbb{P}[x_i \in J \mid (x_i - \boldsymbol{\mu} \prec \boldsymbol{\epsilon}) \text{ and } (\boldsymbol{\mu} - x_i \prec \boldsymbol{\epsilon})] = \gamma$$

**(2)** For examples outside the ROI (far from the mean):

$$\mathbb{P}[x_i \in J \mid (x_i - \boldsymbol{\mu} \nprec \boldsymbol{\epsilon}) \text{ or } (\boldsymbol{\mu} - x_i \nprec \boldsymbol{\epsilon})] = 1 - \gamma$$

Thus, data examples that are closer to the mean are more likely to be included in the training dataset if $\gamma > 0.5$ and in the test dataset otherwise (and vice versa). By varying the bias parameter, $\gamma$, which modulates the distribution shift between the training and test data, we discern a consistent trend favoring the RLP loss across the California Housing, Wine Quality, and Nonlinear datasets. The Nonlinear dataset in particular illustrates that regardless of the selected distribution shift bias, neural networks employing RLP loss invariably outperform, in terms of test MSE, those anchored by MSE loss or its $L_2$ regularized counterpart. These findings emphasize the robustness of RLP loss in the face of distributional disparities between training and test data.

**Noise Scaling Factor.** Given the training dataset, $J$, the objective of this ablation study is to examine the impact of additive Gaussian noise on the performance of RLP loss-trained, MSE loss-trained, and $L_2$ regularized MSE loss-trained models. Specifically, we add standard normal noise scaled by a factor, $\beta$, to each example $x_i \in J$, where $i \in \{1, 2, ..., N\}$. The modified training dataset, $J'$, is denoted

as $J' = \{(x_i', y_i)\}_{i=1}^N$, where:

$$x_i' = x_i + \beta \times \mathcal{N}(\mathbf{0}, \mathbf{I}_d)$$

This experimental setup allows us to gauge how the signal-to-noise ratio (SNR) influences the efficacy of our regression neural network when it is trained using RLP loss, conventional MSE loss, or MSE loss with $L_2$ regularization.

We now evaluate the robustness of the RLP loss under different noise intensities by varying the noise scaling factor, $\beta$. Across the California Housing, Wine Quality, and Nonlinear datasets, for all tested values of $\beta$, the neural network trained using RLP loss consistently achieves a lower test MSE compared to those trained with MSE loss and MSE loss with $L_2$ regularization. Furthermore, as $\beta$ is increased — implying increased noise in the training data — we observe that the RLP loss-trained neural network displays more pronounced asymptotic behavior in the test MSE relative to its counterparts trained with MSE loss and MSE loss with $L_2$ regularization. This behavior indicates that RLP loss not only mitigates the detrimental effects of additive noise but also adapts more effectively to its presence, highlighting its robustness under such data perturbations.

## 5 CONCLUSION

In this work, we presented a new loss function called RLP loss, tailored for capturing non-local linear properties in observed datasets. We provided a mathematical analysis outlining relevant properties of RLP loss, and extended this analysis via rigorous empirical testing on benchmark and synthetic datasets for regression, reconstruction, and classification tasks. It is important to note that training neural networks with RLP loss involves inverting matrices during each training epoch, which is computationally expensive. Optimizing this training process and proving further statistical properties of RLP loss is an open research problem. We consider this work to be a milestone in designing loss functions that capture non-local geometric properties verified by observed datasets.

## Acknowledgements

Shyam Venkatasubramanian and Vahid Tarokh were supported in part by the Air Force Office of Scientific Research (AFOSR) under award FA9550-21-1-0235. Ahmed Aloui was supported in part by the National Science Foundation (NSF) under the National AI Institute for Edge Computing Leveraging Next Generation Wireless Networks Grant # 2112562. Any opinions, findings and conclusions or recommendations expressed in this material are those of the authors and do not necessarily reflect the views of the U.S. Department of Defense and the U.S. National Science Foundation.

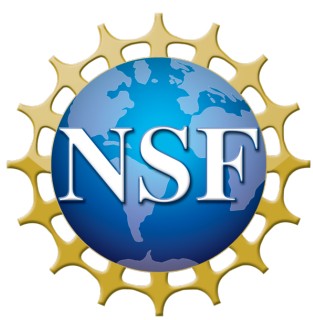

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

# Appendix

## A   PROOFS OF THE THEORETICAL RESULTS

In this section, we present the proofs of the theoretical results outlined in the main text.

*Proposition 2.3.* Let $h \in \mathcal{H}$ be a hypothesis function. We observe that $\mathcal{L}(h) \geq 0$ with the hypothesis minimizing the loss being $h(x) = \mathbb{E}\left[Y|X = x\right]$ almost surely.

*Proof.* Let $\mathbf{X} \in \mathcal{M}_{M,d}(\mathbb{R})$. Firstly, we observe that $\mathcal{L}(h) = \mathbb{E}\left[\left\|\left(\left(\mathbf{X}^{\top}\mathbf{X}\right)^{-1}\mathbf{X}^{\top}\left(\mathbf{Y} - \mathbf{h}(\mathbf{X})\right)\right)^{\top}X\right\|^{2}\right]$ is the expectation of a non-negative random variable. Accordingly, the expectation is non-negative, and therefore, $\mathcal{L}(h) \geqslant 0$.

We start by proving the first implication. Firstly, we suppose that $h(x) = \mathbb{E}\left[Y|X = x\right]$ almost surely. Then, the extension $\mathbf{h}\left(X_1, \ldots, X_M\right) = \left(\mathbb{E}\left[Y|X_1\right], \ldots, \mathbb{E}\left[Y|X_M\right]\right)^{\top}$. Therefore,

$$
\begin{aligned}
\mathcal{L}(h) &= \mathbb{E}\left[\left\|\left(\left(\mathbf{X}^{\top}\mathbf{X}\right)^{-1}\mathbf{X}^{\top}\left(\mathbf{Y} - \mathbf{h}(\mathbf{X})\right)\right)^{\top}X\right\|^{2}\right] \\
&= \mathbb{E}\left[\mathbb{E}\left[\left\|\left(\left(\mathbf{X}^{\top}\mathbf{X}\right)^{-1}\mathbf{X}^{\top}\left(\mathbf{Y} - \mathbf{h}(\mathbf{X})\right)\right)^{\top}X\right\|^{2}\Big|\mathbf{X}\right]\right] \quad \text{(by the law of total expectation)}
\end{aligned}
$$

Let $Z = \left(\left(\mathbf{X}^{\top}\mathbf{X}\right)^{-1}\mathbf{X}^{\top}\left(\mathbf{Y} - \mathbf{h}(\mathbf{X})\right)\right)$, where $Z \in \mathbb{R}^d$. Furthermore, let $X = \left(x_1, \ldots, x_d\right)$, and $Z = \left(z_1, \ldots, z_d\right)$. By linearity of conditional expectation, We have that,

$$
\begin{aligned}
\mathcal{L}(h) &= \mathbb{E}\left[\mathbb{E}\left[\left(\sum_{i=1}^{d} z_i x_i\right)^{2}\Big|\mathbf{X}\right]\right] \\
&= \mathbb{E}\left[\mathbb{E}\left[\sum_{i=1}^{d} z_i^2 x_i^2 + 2\sum_{1 \leq i < j \leq d} z_i z_j x_i x_j \Big|\mathbf{X}\right]\right] \\
&= \mathbb{E}\left[\mathbb{E}\left[\sum_{i=1}^{d} z_i^2 x_i^2 \Big|\mathbf{X}\right] + 2\,\mathbb{E}\left[\sum_{1 \leq i < j \leq d} z_i z_j x_i x_j \Big|\mathbf{X}\right]\right] \\
&= \mathbb{E}\left[\sum_{i=1}^{d} \mathbb{E}\left[z_i^2 x_i^2 \Big|\mathbf{X}\right] + 2\sum_{1 \leq i < j \leq d} \mathbb{E}\left[z_i z_j x_i x_j \Big|\mathbf{X}\right]\right] \\
&= \mathbb{E}\left[\sum_{i=1}^{d} \mathbb{E}\left[z_i^2|\mathbf{X}\right]\mathbb{E}\left[x_i^2\right] + 2\sum_{1 \leq i < j \leq d} \mathbb{E}\left[z_i|\mathbf{X}\right]\mathbb{E}\left[z_j|\mathbf{X}\right]\mathbb{E}\left[x_i\right]\mathbb{E}\left[x_j\right]\right]
\end{aligned}
$$

The last equation follows from the following independence conditions: $X \perp\!\!\!\perp \mathbf{X}$, $Z \perp\!\!\!\perp X|\mathbf{X}$, and $z_i \perp\!\!\!\perp z_j|\mathbf{X}$.

We now prove that for every $i \in \{1, \ldots, d\}$, $\mathbb{E}[z_i^2|\mathbf{X}] = 0$ and $\mathbb{E}[z_i|\mathbf{X}] = 0$.

Let $\mathbf{A} = \left(\mathbf{X}^{\top}\mathbf{X}\right)^{-1}\mathbf{X}^{\top}$, where $\mathbf{A} = \begin{pmatrix} a_{11} & \cdots & a_{1M} \\ \vdots & \ddots & \vdots \\ a_{d1} & \cdots & a_{dM} \end{pmatrix}$. We have that $z_i = \sum_{k=1}^{M} a_{ik}\left(Y_k - h(X_k)\right)$.

Therefore, by linearity of the conditional expectation, and since we considered that $h = \mathbb{E}[Y_k|X_k]$, we have $\mathbb{E}[z_i|\mathbf{X}] = 0$.

Furthermore, we have that,

$$\mathbb{E}[z_i^2|\mathbf{X}] = \mathbb{E}\left[\left(\sum_{k=1}^{M} a_{ik}\left(Y_k - h(X_k)\right)\right)^2 \Bigg|\mathbf{X}\right]$$

$$= \mathbb{E}\left[\sum_{k=1}^{M} a_{ik}^2\left(Y_k - h(X_k)\right)^2 \Bigg|\mathbf{X}\right] + 2\,\mathbb{E}\left[\sum_{1 \le k < l \le M} a_{ik}a_{il}\left(Y_k - h(X_k)\right)\left(Y_l - h(X_l)\right)\Bigg|\mathbf{X}\right]$$

$$= \sum_{k=1}^{M} \mathbb{E}\left[a_{ik}^2\Big|\mathbf{X}\right]\mathbb{E}\left[\left(Y_k - h(X_k)\right)^2\Big|\mathbf{X}\right] + 2\sum_{k<l}\mathbb{E}\left[a_{ik}a_{il}\Big|\mathbf{X}\right]\mathbb{E}\left[\left(Y_k - h(X_k)\right)\Big|\mathbf{X}\right]\mathbb{E}\left[\left(Y_l - h(X_l)\right)\Big|\mathbf{X}\right]$$

$$= 0$$

We now prove the second implication, assuming that $\mathcal{L}(h) = 0$. Finding the minimum over the space of functions, $\mathcal{H}$, is equivalent to solving for $\mathbf{h}(x)$ for every $x$. Subsequently, letting $\mathbf{x} \in \mathcal{M}_{M,d}(\mathbb{R})$, we have that,

$$\mathbf{h}(\mathbf{x}) = \arg\min_{\mathbf{c}\in\mathbb{R}^M} \mathbb{E}\left[\left\|X^\top\left(\left(\mathbf{X}^\top\mathbf{X}\right)^{-1}\mathbf{X}^\top\left(\mathbf{Y}-\mathbf{c}\right)\right)\right\|^2 \Bigg|\mathbf{X}=\mathbf{x}\right]$$

By taking the gradient with respect to $\mathbf{c}$, we have that,

$$\nabla_{\mathbf{c}}\mathbb{E}\left[\left\|X^\top\left(\left(\mathbf{X}^\top\mathbf{X}\right)^{-1}\mathbf{X}^\top\left(\mathbf{Y}-\mathbf{c}\right)\right)\right\|^2 \Bigg|\mathbf{X}=\mathbf{x}\right] = \mathbb{E}\left[\nabla_{\mathbf{c}}\left\|X^\top\left(\left(\mathbf{X}^\top\mathbf{X}\right)^{-1}\mathbf{X}^\top\left(\mathbf{Y}-\mathbf{c}\right)\right)\right\|^2 \Bigg|\mathbf{X}=\mathbf{x}\right]$$

$$= \mathbb{E}\left[\left(X^\top\mathbf{A}\left(\mathbf{Y}-\mathbf{c}\right)\right)\left(\left(\mathbf{Y}-\mathbf{c}\right)\odot\mathbf{A}^\top X\right)\Bigg|\mathbf{X}=\mathbf{x}\right]$$

Consequently, if the gradient with respect to $\mathbf{c}$ is zero, it implies that for every $i \in \{1,\ldots,M\}$, we have that,

$$\sum_{j=1}^{d}\sum_{k=1}^{M}\sum_{l=1}^{d}\mathbb{E}\left[x_l x_j a_{jk} a_{li}\left(y_k - c_k\right)\left(y_i - c_i\right)\Big|\mathbf{X}=\mathbf{x}\right] = 0$$

where $x_i$ is the $i^{th}$ component of $X$ and $a_{jk}$ are the elements of the matrix $\mathbf{A}$. By the independence of $X$, and the fact that $a_{jk}$ is $\mathbf{X}$-measurable, it follows that if the gradient is zero. Accordingly, we have that,

$$\sum_{j=1}^{d}\sum_{k=1}^{M}\sum_{l=1}^{d}\mathbb{E}\left[x_l x_j\right] a_{jk} a_{li}\mathbb{E}\left[\left(y_k - c_k\right)\left(y_i - c_i\right)\Big|\mathbf{X}=\mathbf{x}\right] = 0$$

Since the rows of $\mathbf{X}$ are independent and identically distributed and since $M > d$, we have that $\mathbf{X}^\top\mathbf{X}$ is full rank and invertible, and hence, $\mathbf{A}$ is positive definite. Furthermore, $\mathbb{E}\left[x_l x_j\right]$ are the elements of the covariance matrix of $X$, which is also positive definite. If the gradient is equal to zero, this implies that, for every $i, k \in \{1,\ldots,M\}$,

$$\mathbb{E}\left[\left(y_k - c_k\right)\left(y_i - c_i\right)\Big|\mathbf{X}=\mathbf{x}\right] = 0$$

Consequently, for $i = k$,

$$\mathbb{E}\left[\left(y_i - c_i\right)^2\Big|\mathbf{X}=\mathbf{x}\right] = 0$$

Hence,

$$\mathbf{c} = \mathbb{E}\left[\mathbf{Y}\big|\mathbf{X}=\mathbf{x}\right]$$

Therefore, we see that $\mathcal{L}(h) \ge 0$ with the hypothesis minimizing the loss being $h(x) = \mathbb{E}\left[Y|X=x\right]$ almost surely. $\qquad\square$

*Proposition 2.4.* Let $L_0$ denote the MSE loss and let $\theta^*$ be the optimal parameters (i.e $h_{\theta^*} = \mathbb{E}\left[Y|X\right]$ almost surely). We assume that both the MSE and RLP loss functions are convex. Under the following conditions:

(i) $\mathbb{E}\left[X_i X_j\right] = [1, \cdots, 1]^\top \mathbb{1}_{i=j}$.

(ii) $(\mathbf{Y} - h_\theta(\mathbf{X})) \leqslant 0$ and $\nabla_\theta h_\theta(\mathbf{X}) \leqslant 0$ (component-wise inequality).

(iii) For every $j, k \in \{1, 2, \ldots, d\}$ and for every $l \in \{1, 2, \ldots, M\}$, $\mathbb{E}[\mathbf{a}_{jk}\mathbf{a}_{kl}] \geqslant \frac{1}{d^2}$, where $(\mathbf{a}_{jk})$ and $(\mathbf{a}_{kl})$ are the components of $\mathbf{A} = (\mathbf{X}^\top \mathbf{X})^{-1}\mathbf{X}^\top$.

We observe that for every step size $\epsilon \geqslant 0$ and parameter $\theta \in \mathbb{R}^W$ for which gradient descent converges,

$$\|\theta^* - (\theta - \epsilon \nabla_\theta \mathcal{L}(\theta))\| \leqslant \|\theta^* - (\theta - \epsilon \nabla_\theta L_0(\theta))\|$$

This proposition contrasts the convergence behavior of the two loss functions, MSE loss and RLP loss, for gradient descent optimization in parameterized models. It asserts that under certain conditions — (*i*), (*ii*), and (*iii*) from Proposition 2.4 — updates based on the gradient of the RLP loss function bring the parameters closer to the optimal solution than those based on the gradient of the MSE loss function.

*Proof.* Under the following assumptions: (*i*) $\mathbb{E}[X_i X_j] = [1, \cdots, 1]^\top \mathbb{1}_{i=j}$.

(*ii*) $(\mathbf{Y} - h_\theta(\mathbf{X})) \leqslant 0$ and $\nabla_\theta h_\theta(\mathbf{X}) \leqslant 0$ (component-wise inequality)

(*iii*) $E[\mathbf{a}_{jk}\mathbf{a}_{lk}] \geqslant \frac{1}{d^2} \forall j, k, l$, where $(\mathbf{a}_{im})_{1 \leq i \leq d, 1 \leq m \leq M}$ are the components of the matrix $\mathbf{A} = (\mathbf{X}^\top \mathbf{X})^{-1}\mathbf{X}^\top$

We have that,

$$\|\theta^* - \theta + \varepsilon \nabla_\theta \mathcal{L}(\theta)\|_2^2 = \sum_{i=1}^{K} \left(\theta_i^* - \theta_i + \epsilon \frac{\partial}{\partial \theta_i}\mathcal{L}(\theta)\right)^2$$

Letting $1 \leqslant i \leqslant W$, we have that,

$$\frac{\partial}{\partial \theta_i}\mathcal{L}(\theta) = -2\,\mathbb{E}\left[X_{n+1}^\top \left((\mathbf{X}^\top \mathbf{X})^{-1}\mathbf{X}^\top (\mathbf{Y} - h(\mathbf{X}))\right) \times X_{n+1}^\top \left((\mathbf{X}^\top \mathbf{X})^{-1}\mathbf{X}^\top \frac{\partial}{\partial \theta_i}h(\mathbf{X})\right)\right]$$

$$= -2\,\mathbb{E}\left[X_{n+1}^2\right]^\top \mathbb{E}\left[\left((\mathbf{X}^\top \mathbf{X})^{-1}\mathbf{X}^\top (\mathbf{Y} - h(\mathbf{X}))\right) \odot \left((\mathbf{X}^\top \mathbf{X})^{-1}\mathbf{X}^\top \frac{\partial}{\partial \theta_i}h(\mathbf{X})\right)\right]$$

Where $\odot$ denotes the Hadamard product between two vectors. Note that $\frac{1}{2}\frac{\partial}{\partial \theta_i}\mathcal{L}(\theta) \leqslant 0$. Subsequently, it follows from assumption *(i)* that,

$$-\frac{1}{2}\frac{\partial}{\partial \theta_i}\mathcal{L}(\theta) = \mathbb{E}\left[(\mathbf{Y} - h(\mathbf{X}))^\top \mathbf{A}^\top \mathbf{A} \frac{\partial}{\partial \theta_i}h_\theta(\mathbf{X})\right]$$

$$= \sum_{j=1}^{M}\sum_{l=1}^{M}\sum_{k=1}^{d} \mathbb{E}\left[(Y_j - h(X_j))\,\mathbf{a}_{jk}\mathbf{a}_{lk}\frac{\partial}{\partial \theta_i}h(X_l)\right]$$

$$= \sum_{j=1}^{M}\sum_{l=1}^{M}\sum_{k=1}^{d} \mathbb{E}\left[\mathbb{E}\left[(Y_j - h(X_j))\,\mathbf{a}_{jk}\mathbf{a}_{lk}\frac{\partial}{\partial \theta_i}h(X_l)\right]\bigg|X_j, X_l\right]$$

$$= \sum_{k=1}^{d}\sum_{j \neq l}^{M} \mathbb{E}[(Y_j - h(X_j))]\,\mathbb{E}\left[\frac{\partial}{\partial \theta_i}h(X_l)\right]\mathbb{E}[\mathbf{a}_{jk}\mathbf{a}_{lk}]$$

$$+ \sum_{k=1}^{d}\sum_{j=1}^{M} \mathbb{E}\left[(Y_j - h(X_j))\frac{\partial}{\partial \theta_i}h(X_j)\right]\mathbb{E}[\mathbf{a}_{jk}\mathbf{a}_{jk}]$$

$$\geqslant \frac{M}{d}\,\mathbb{E}\left[(Y_1 - h(X_1))\frac{\partial}{\partial \theta_i}h(X_1)\right]$$

$$\geqslant -\frac{1}{2}\frac{\partial}{\partial \theta_i}L_0(\theta)$$

This result follows from the application of the tower property, noting that for $j \neq l$, we have that $Y_j \perp\!\!\!\perp Y_l | X_j, X_l$ and $h(X_j) \perp\!\!\!\perp h(X)_l | X_j, X_l$, and by applying assumption *(ii)* and *(iii)*. Therefore we have that,

$$\left(\theta_i^* - \theta_i + \epsilon \frac{\partial}{\partial \theta_i}\mathcal{L}(\theta)\right)^2 \leqslant \left(\theta_i^* - \theta_i + \epsilon \frac{\partial}{\partial \theta_i}L_0(\theta)\right)^2$$

Accordingly, we observe that $\|\theta^* - (\theta - \epsilon\nabla_\theta\mathcal{L}(\theta))\| \leqslant \|\theta^* - (\theta - \epsilon\nabla_\theta L_0(\theta))\|$ for every step size $\epsilon \geqslant 0$ and parameter $\theta \in \mathbb{R}^W$ for which gradient descent converges. $\qquad\square$

# B  DATASET DESCRIPTIONS

## B.1  CALIFORNIA HOUSING DATASET

The **California Housing** dataset contains housing data of California derived from the 1990 U.S. census. It is often used for regression predictive modeling tasks. The dataset has:

- $|\mathcal{X}| = 20640$ examples, with $|J|$ training examples and $|\mathcal{X}| - |J|$ test examples.
- $d = 8$ features: MedInc, HouseAge, AveRooms, AveBedrms, Population, AveOccup, Latitude, and Longitude.
- Target Variable: Median house value for California districts.

## B.2  WINE QUALITY DATASET

The **Wine Quality** dataset from consists of physicochemical tests and the quality of red and white vinho verde wine samples, from the north of Portugal. The dataset has:

- $|\mathcal{X}| = 6497$ examples (combined red and white wine), with $|J|$ training examples and $|\mathcal{X}| - |J|$ test examples.
- $d = 11$ features: fixed acidity, volatile acidity, citric acid, residual sugar, chlorides, free sulfur dioxide, total sulfur dioxide, density, pH, sulphates, and alcohol.
- Target Variable: Quality score between 0 and 10.

## B.3  LINEAR SYNTHETIC DATASET

The **Linear** dataset is a synthetic dataset, generated with fixed random seed, `rng = np.random.RandomState(0)` (in python code). The dataset has:

- $|\mathcal{X}| = 6000$ examples, with $|J|$ training examples and $|\mathcal{X}| - |J|$ test examples.
- $d = 5$ features: $\mathcal{X}_1, \mathcal{X}_2, \mathcal{X}_3, \mathcal{X}_4, \mathcal{X}_5 \sim \mathcal{U}[0, 1]$, where each feature is uniformly distributed between 0 and 1.
- Target Variable: Given by the equation

$$\mathcal{Y} = 0.5\mathcal{X}_1 + 1.5\mathcal{X}_2 + 2.5\mathcal{X}_3 + 3.5\mathcal{X}_4 + 4.5\mathcal{X}_5$$

## B.4  NONLINEAR SYNTHETIC DATASET

The **Nonlinear** dataset is a synthetic dataset, produced with fixed random seed, `rng = np.random.RandomState(1)` (in python code). The dataset has:

- $|\mathcal{X}| = 6000$ examples, with $|J|$ training examples and $|\mathcal{X}| - |J|$ test examples.
- $d = 7$ features: $\mathcal{X}_1, \mathcal{X}_2, \mathcal{X}_3, \mathcal{X}_4, \mathcal{X}_5, \mathcal{X}_6, \mathcal{X}_7 \sim \mathcal{U}[0, 1]$, where each feature is uniformly distributed between 0 and 1.
- Target Variable: Given by the equation

$$\mathcal{Y} = \mathcal{X}_1 + \mathcal{X}_2^2 + \mathcal{X}_3^3 + \mathcal{X}_4^4 + \mathcal{X}_5^5 + e^{\mathcal{X}_6} + \sin(\mathcal{X}_7)$$

### B.5 MNIST DATASET

The **MNIST** (Modified National Institute of Standards and Technology) dataset is a collection of handwritten digits commonly used for training image processing systems. While the original MNIST dataset consists of 50000 training and 10000 test examples, we consider a smaller version of the dataset (randomly partitioned from the original training and test datasets) that has:

- $|\mathcal{X}| = 10000$ examples, with $|J|$ training examples (from the MNIST training examples) and $|\mathcal{X}| - |J|$ test examples (from the MNIST test examples).
- Each example (image) is of size $28 \times 28$ pixels, represented as a grayscale intensity from 0 to 255.
- Target Variable: The actual digit the image represents, ranging from 0 to 9.

### B.6 CIFAR-10 DATASET

The **CIFAR-10** dataset comprises color images categorized into 10 different classes, representing various objects and animals such as airplanes, cars, and birds. The images cover a broad range of scenarios, making the dataset highly versatile for various computer vision tasks. While the original CIFAR-10 dataset consists of 50000 training and 10000 test examples, we consider a smaller version of the dataset (randomly partitioned from the original training and test datasets) that has:

- $|\mathcal{X}| = 10000$ examples, with $|J|$ training examples (from the CIFAR-10 training examples) and $|\mathcal{X}| - |J|$ test examples (from the CIFAR-10 test examples).
- Each example (image) is of size $32 \times 32 \times 3$, with three color channels (Red, Green, Blue), and size $32 \times 32$ pixels for each channel, represented as a grayscale intensity from 0 to 255.
- Target Variable: The class label of the image.

## C ADDITIONAL EXPERIMENTS

### C.1 CLASSIFICATION TASKS

While the RLP loss was introduced in the scope of regression and reconstruction tasks, we note that the loss can also be applied to classification tasks. We provide a motivation for using the RLP loss for classification in Figure 8 — paralleling the regression case, we note that if two discontinuous functions with discrete images share the same hyperplanes connecting all subsets of their feature-label pairs, then they must necessarily be equivalent.

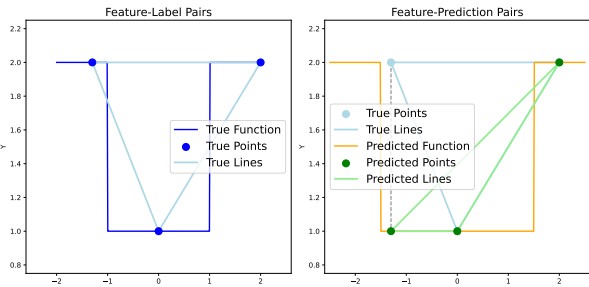

Figure 8: Comparison of true and predicted functions — a demonstration that two discontinuous functions with discrete images are equivalent if and only if they share identical hyperplanes generated by all possible feature-label pairs.

Accordingly, we observe that minimizing the RLP loss (and achieving zero loss) ensures that we learn the true [discontinuous] function — this is supported by our theoretical findings in Section A. Our empirical results, obtained from datasets such as the Moons dataset (`sklearn.datasets.make_moons` in python) and MNIST, affirm that the RLP loss offers accelerated convergence and superior outcomes in terms of accuracy and the $F_1$-score. Additionally, we employ mixup [Zhang et al., 2017] and juxtapose RLP loss against the cross-entropy loss when both are combined with mixup data augmentation (we further investigate mixup data augmentation for regression in section C.2). The results are illustrated in Figures 9 and 10.

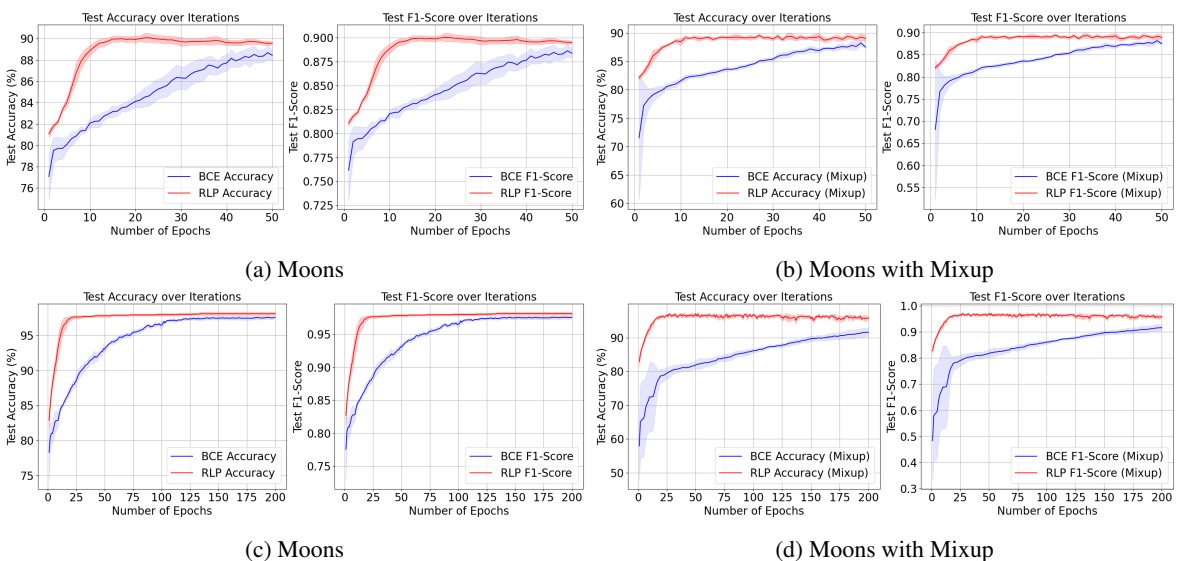

Figure 9: Comparing performance between Cross Entropy Loss and Random Linear Projections Loss for a classification task on the Moons dataset in terms of accuracy and $F_1$-score. Figure 9a showcases results with $|J| = 900$ training and $|\mathcal{X}| - |J| = 100$ test examples ($|\mathcal{X}| = 1000$). Figure 9b uses the same data split but is augmented with mixup. Figure 9c employs a smaller set of $|J| = 25$ training examples and $|\mathcal{X}| - |J| = 475$ test examples ($|\mathcal{X}| = 500$), while Figure 9d integrates the mixup data augmentation method on this smaller dataset. Both loss functions are evaluated across all scenarios.

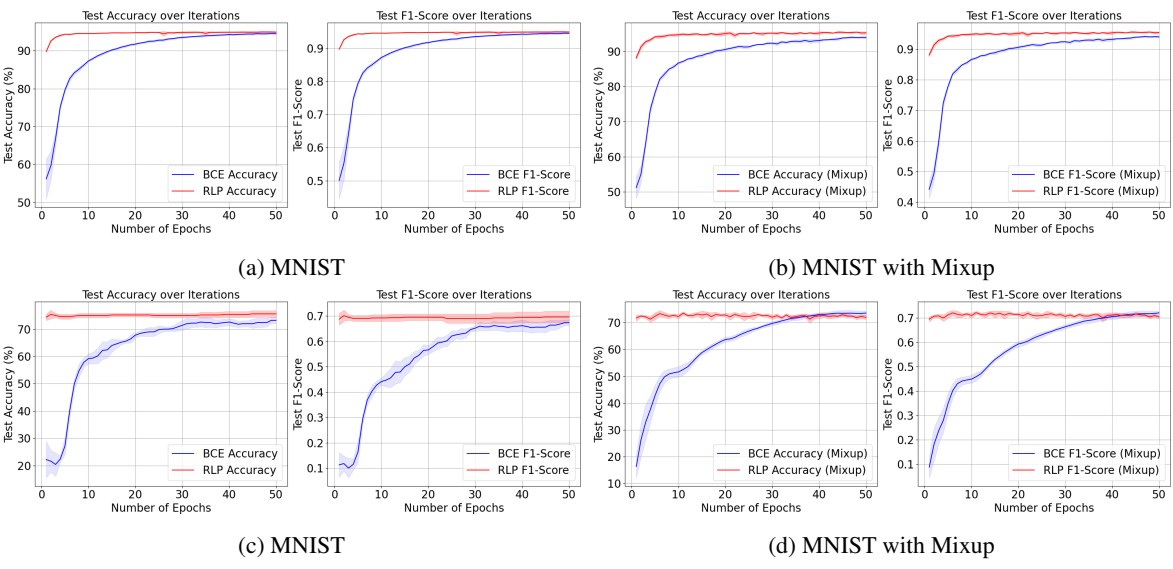

Figure 10: Performance comparison between Cross Entropy Loss and Random Linear Projections Loss for a classification task on MNIST, evaluated in terms of accuracy and $F_1$-score. Figure 10a showcases results with $|J| = 5000$ training and $|X| - |J| = 5000$ test examples ($|X| = 10000$). Figure 10b uses the same data split but is augmented with the mixup method. Figure 10c employs a smaller set of $|J| = 100$ training and $|X| - |J| = 1000$ test examples, while Figure 10d integrates the mixup data augmentation method on this smaller dataset. Both loss functions are evaluated across all scenarios.

## C.2 REGRESSION AND RECONSTRUCTION TASKS

We now provide additional empirical results pertaining to the regression and reconstruction tasks outlined in the main text. As an extension, we compare RLP loss with **(1)** mixup-augmented MSE loss (MSE loss + Mixup) and **(2)** mixup-augmented RLP loss (RLP loss + Mixup). Regarding **(1)**, we use MSE loss to train the neural network on the virtual training examples produced by mixup, whereas in **(2)**, we use RLP loss to train the neural network on the virtual training examples formed

using convex combinations between two unique pairs of sets of hyperplanes connecting fixed-size subsets of the neural network's feature-prediction pairs and feature-label pairs (see Algorithm 3).

---

**Algorithm 3:** Neural Network Training With Mixup-Augmented RLP Loss

---

**Input:** $J$ (Training dataset), $\theta$ (Initial NN parameters), $\alpha$ (Learning rate), $M$ (Batch size),
$\quad\quad K$ (Number of batches to generate), $E$ (Number of epochs), $\psi$ (Beta distribution shape parameter)
**Output:** $\theta$ (Trained NN parameters)

**1** $\mathcal{B}_a \leftarrow$ `Balanced_Batch_Generator`$(J, M, K)$
**2** $\mathcal{B}_b \leftarrow$ `Balanced_Batch_Generator`$(J, M, K)$
**3 for** $epoch = 1, 2, \ldots, E$ **do**
**4** $\quad$ **for** $j = 1, 2, \ldots, K$ **do**
**5** $\quad\quad$ $\mathbf{x}_a, \mathbf{x}_b \leftarrow$ Matrix of features from batches $\mathcal{B}_a[j]$ and $\mathcal{B}_b[j]$, respectively
**6** $\quad\quad$ $\mathbf{y}_a, \mathbf{y}_b \leftarrow$ Vector of labels from batches $\mathcal{B}_a[j]$ and $\mathcal{B}_b[j]$, respectively
**7** $\quad\quad$ **if** $\text{size}(\mathbf{x}_a) \neq \text{size}(\mathbf{x}_b)$ **then**
**8** $\quad\quad$ $\lambda \leftarrow \text{Beta}(\psi, \psi)$ (Randomly sample from Beta distribution)
**9** $\quad\quad$ $\mathbf{x}_j \leftarrow (\lambda)\mathbf{x}_a + (1 - \lambda)\mathbf{x}_b$
**10** $\quad\quad$ $\mathbf{y}_j \leftarrow (\lambda)\mathbf{y}_a + (1 - \lambda)\mathbf{y}_b$
**11** $\quad\quad$ $M_y \leftarrow (\mathbf{x}_j^\top \mathbf{x}_j)^{-1} \mathbf{x}_j^\top \mathbf{y}_j$
**12** $\quad\quad$ $M_h \leftarrow (\mathbf{x}_j^\top \mathbf{x}_j)^{-1} \mathbf{x}_j^\top \mathbf{h}_\theta(\mathbf{x}_j)$
**13** $\quad\quad$ $l_j(\theta) \leftarrow \left( \sum_{k=1}^{M} (M_y - M_h)^\top x_{j_k} \right)^2$
**14** $\quad$ $\mathcal{L}(\theta) \leftarrow \frac{1}{K} \sum_{j=1}^{K} l_j(\theta)$
**15** $\quad$ $\theta \leftarrow \theta - \alpha \nabla_\theta \mathcal{L}(\theta)$
**16 return** $\theta$

---

### C.2.1 Performance Analysis

Extending the first evaluation from the main text, we evaluate the efficacy of RLP loss compared to the mixup-augmented MSE loss and mixup-augmented RLP loss, when there are no ablations introduced within the data. We observe that across all three datasets, neural networks trained with RLP loss and mixup-augmented RLP loss achieve improved performance when compared to those trained with mixup-augmented MSE loss. The results are illustrated in Figure 11.

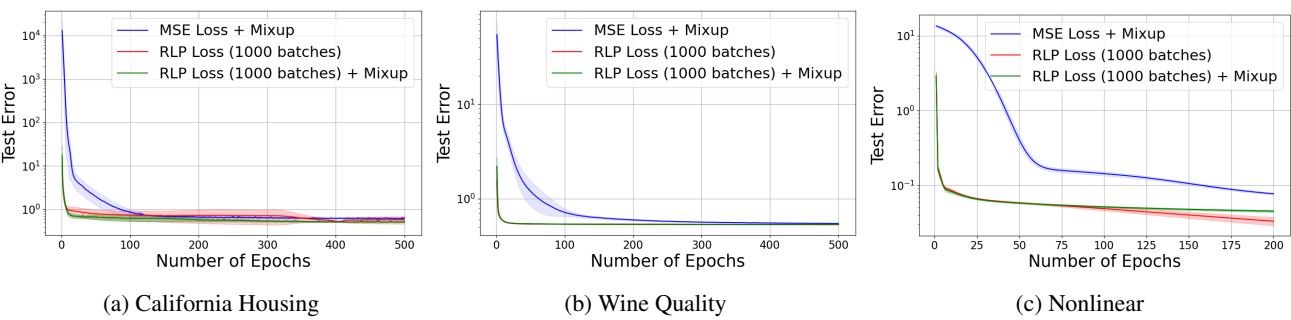

(a) California Housing $\quad\quad\quad\quad$ (b) Wine Quality $\quad\quad\quad\quad$ (c) Nonlinear

Figure 11: Test performance comparison across three datasets (California Housing, Wine Quality, and Nonlinear) using three different loss functions: mixup-augmented MSE, RLP, and mixup-augmented RLP. The x-axis represents training epochs, while the y-axis indicates the test MSE.

We also compare the elapsed training time (in seconds) using RLP loss versus MSE loss on the California Housing, Wine Quality, and Nonlinear datasets. This result was compiled with an Intel Xeon CPU @ 2.20GHz, and is depicted in Figure 12. For the RLP loss case, we train the neural network using $K = 2000$ batches.

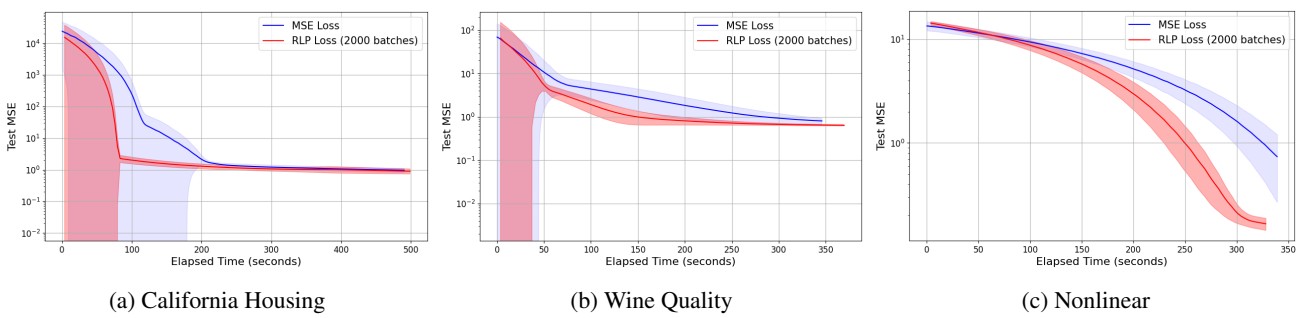

(a) California Housing  (b) Wine Quality  (c) Nonlinear

Figure 12: Test performance comparison across three datasets (California Housing, Wine Quality, and Nonlinear) using MSE loss and RLP loss. The x-axis represents elapsed time in seconds, while the y-axis indicates the test MSE.

### C.2.2 Ablation Study — Number of Training Examples

For the ablation study pertaining to the number of training examples, $|J|$, we first consider the case where $|J| = 100$ training examples. For this case, we train the regression neural network using MSE loss, MSE loss with $L_2$ regularization, or RLP loss. As in the $|J| = 50$ case from the main text, we observe that the RLP loss-trained models consistently outperform their MSE loss and $L_2$ regularized MSE loss-trained counterparts in both convergence rate and test error, despite the limited data. These results are illustrated in Figure 13.

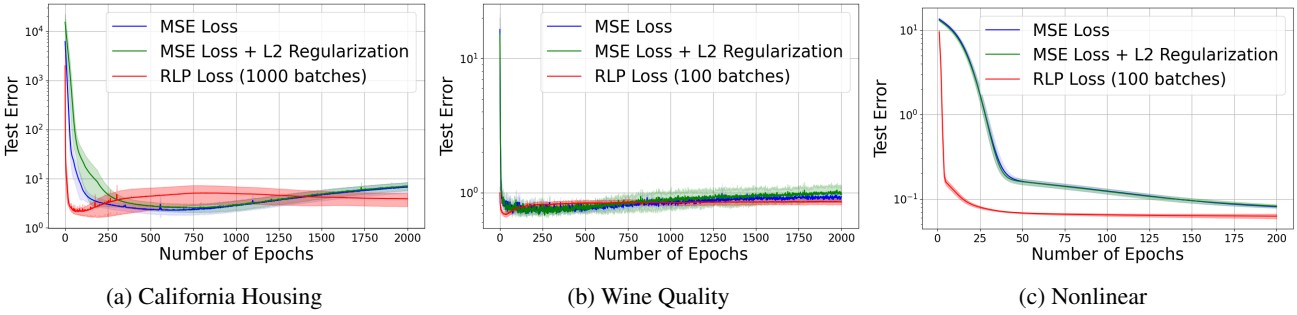

(a) California Housing  (b) Wine Quality  (c) Nonlinear

Figure 13: Limited training data ($|J| = 100$) test performance comparison across three datasets (California Housing, Wine Quality, and Nonlinear) using three different loss functions: MSE, MSE with $L_2$ regularization (MSE + $L_2$), and RLP. The x-axis represents training epochs, while the y-axis indicates the test MSE.

We also consider the case where $|J| = 100$ for the image reconstruction task. We observe that across both CIFAR-10 and MNIST, neural networks trained with RLP loss achieve improved performance when compared to those trained with MSE loss and $L_2$ regularized MSE loss. These results are illustrated in Figure 14.

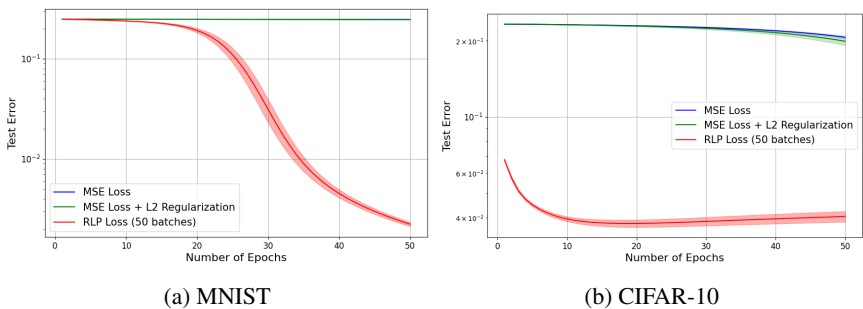

(a) MNIST  (b) CIFAR-10

Figure 14: Limited training data ($|J| = 100$) test performance comparison across two datasets (CIFAR-10 and MNIST) using three different loss functions: MSE, MSE with $L_2$ regularization, and RLP. The x-axis represents training epochs, while the y-axis indicates the test MSE.

Subsequently, we extend this study by evaluating the efficacy of RLP loss compared to the mixup-augmented MSE loss and

mixup-augmented RLP loss when $|J| \in \{50, 100\}$. We see that across all three datasets (California Housing, Wine Quality, and Nonlinear), neural networks trained with RLP loss and mixup-augmented RLP loss achieve improved performance when compared to those trained with mixup-augmented MSE loss. These results are illustrated in Figures 15 and 16.

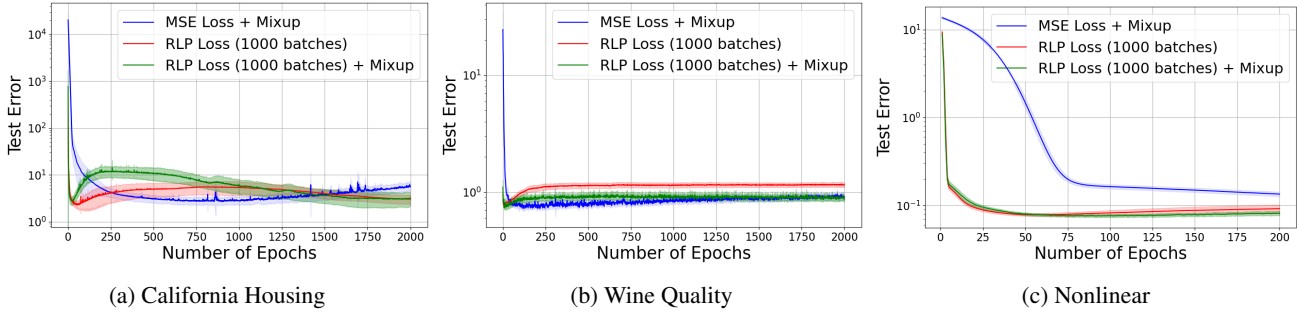

(a) California Housing        (b) Wine Quality        (c) Nonlinear

Figure 15: Limited training data ($|J| = 50$) test performance comparison across three datasets (California Housing, Wine Quality, and Nonlinear) using three different loss functions: mixup-augmented MSE, RLP, and mixup-augmented RLP. The x-axis represents training epochs, while the y-axis indicates the test MSE.

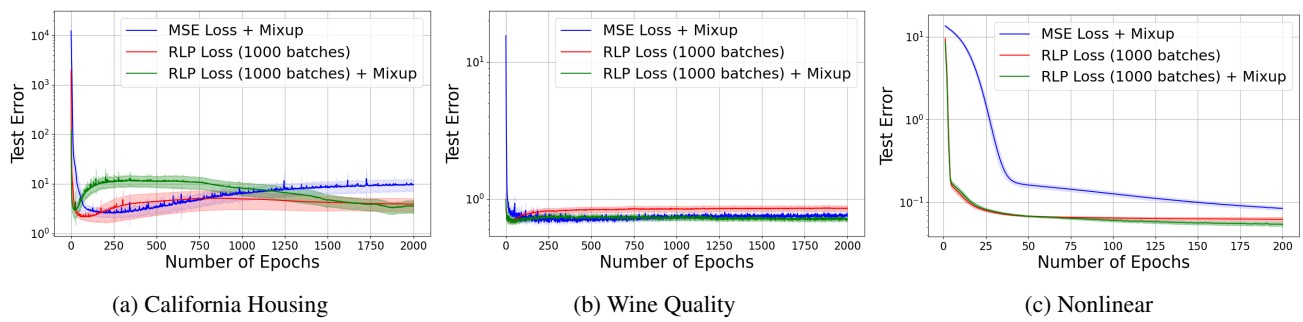

(a) California Housing        (b) Wine Quality        (c) Nonlinear

Figure 16: Limited training data ($|J| = 100$) test performance comparison across three datasets (California Housing, Wine Quality, and Nonlinear) using three different loss functions: mixup-augmented MSE, RLP, and mixup-augmented RLP. The x-axis represents training epochs, while the y-axis indicates the test MSE.

### C.2.3 Ablation Study — Distribution Shift Bias

We extend the distribution shift bias ablation study by evaluating the efficacy of RLP loss compared to the mixup-augmented MSE loss and mixup-augmented RLP loss for bias parameter $\gamma \in \{0.1, 0.2, \ldots, 0.9\}$. We observe that across all three datasets, neural networks trained with RLP loss and mixup-augmented RLP loss achieve improved performance when compared to those trained with mixup-augmented MSE loss. This result is illustrated in Figure 17.

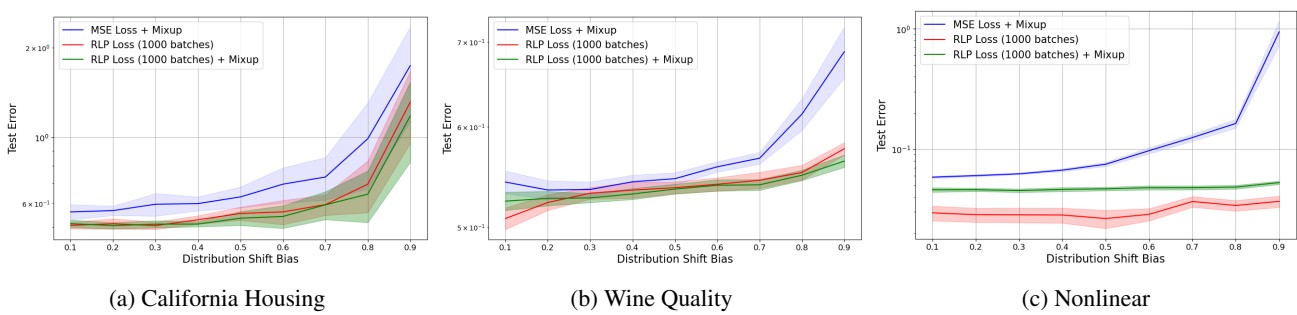

(a) California Housing        (b) Wine Quality        (c) Nonlinear

Figure 17: Distribution shift test performance comparison across three datasets (California Housing, Wine Quality, and Nonlinear) using three different loss functions: mixup-augmented MSE, RLP, and mixup-augmented RLP. The x-axis is the degree of bias, $\gamma$, between the test data and the training data, while the y-axis indicates the test MSE.

### C.2.4 Ablation Study — Noise Scaling Factor

We extend the noise scaling factor ablation study by evaluating the efficacy of RLP loss compared to the mixup-augmented MSE loss and mixup-augmented RLP loss for standard normal noise scaling factor $\beta \in \{0.1, 0.2, \ldots, 0.9\}$. We observe that across all three datasets, neural networks trained with RLP loss and mixup-augmented RLP loss achieve improved performance when compared to those trained with mixup-augmented MSE loss. This result is illustrated in Figure 18.

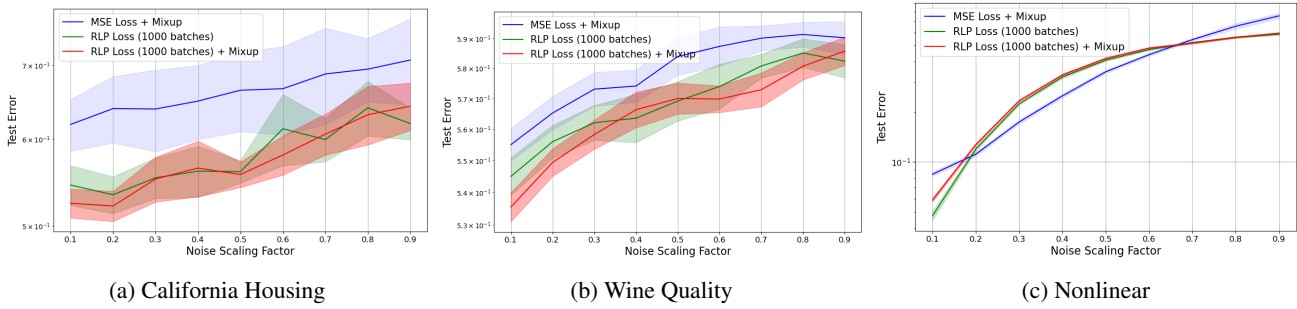

| (a) California Housing | (b) Wine Quality | (c) Nonlinear |

Figure 18: Noise robustness test performance comparison across three datasets (California Housing, Wine Quality, and Nonlinear) using three different loss functions: mixup-augmented MSE, RLP, and mixup-augmented RLP. The x-axis is the scaling factor, $\beta$, for the additive standard normal noise, while the y-axis indicates the test MSE.

### C.2.5 RLP Loss vs. Mean Absolute Error (MAE) Loss

Mean Absolute Error (MAE) loss is widely utilized in machine learning for its simplicity and interpretability, particularly in regression tasks. Its effectiveness is underscored by research demonstrating its superiority in vector-to-vector regression and in enhancing neural network training with noisy labels, showcasing its adaptability across various applications Qi et al. [2020], Zhang and Sabuncu [2018]. Paralleling the first evaluation from the main text, we evaluate the efficacy of RLP loss compared to MAE loss, when there are no ablations introduced within the data, for $|J| = 0.5|\mathcal{X}|$ training examples and $|\mathcal{X}| - |J|$ test examples. We observe that across all three datasets, neural networks trained with RLP loss achieve improved performance when compared to those trained with MAE loss. The results are illustrated in Figure 19.

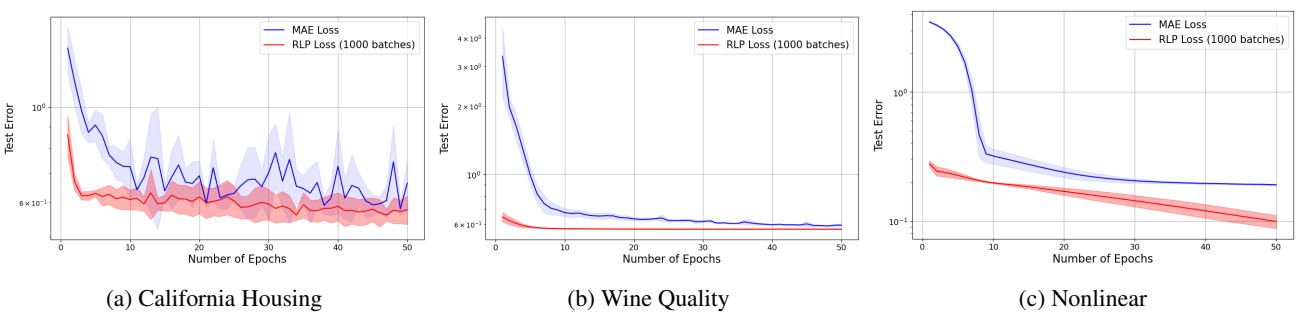

| (a) California Housing | (b) Wine Quality | (c) Nonlinear |

Figure 19: Test performance comparison across three datasets (California Housing, Wine Quality, and Nonlinear) using two different loss functions: RLP and MAE. The x-axis represents training epochs, while the y-axis indicates the test MAE.

## D EXPERIMENT DETAILS

### D.1 NEURAL NETWORK ARCHITECTURES

We first provide a detailed description of four different neural network architectures designed for various tasks: regression, image reconstruction, and classification. Each of these architectures were employed to generate the respective empirical results pertaining to the aforementioned tasks.

### D.1.1 Regression Neural Network

The Regression Neural Network utilized in our analysis is designed for regression tasks, mapping input features to continuous output values (see Figure 20). The architecture comprises the following layers:

- **Fully Connected Layer (`fc1`)**: Transforms the input features to a higher dimensional space. It takes $d$-dimensional inputs and yields 32-dimensional outputs.
- **ReLU Activation (`relu1`)**: Introduces non-linearity to the model. It operates element-wise on the output of `fc1`.
- **Fully Connected Layer (`fc2`)**: Takes 32-dimensional inputs and yields 1-dimensional outputs (final predictions).

### D.1.2 Autoencoders for Image Reconstruction

The Autoencoder utilized in our analysis is tailored for image reconstruction tasks (see Figure 20). The architecture consists of two main parts: an encoder and a decoder. We note that preliminarily, all images are flattened to $d$-dimensional inputs and have their pixel values normalized to be within the range $[0, 1]$.

- **Encoder**:
  - **Fully Connected Layer (`fc1`)**: Encodes the flattened, $d$-dimensional input into a latent representation of size 32.
  - **ReLU Activation (`relu1`)**: Introduces non-linearity to the encoding process.
- **Decoder**:
  - **Fully Connected Layer (`fc2`)**: Transforms the 32-dimensional latent representation into a $d$-dimensional output.
  - **Sigmoid Activation (`sig1`)**: Ensures the output values are in the range $[0, 1]$ (akin to normalized pixel values).

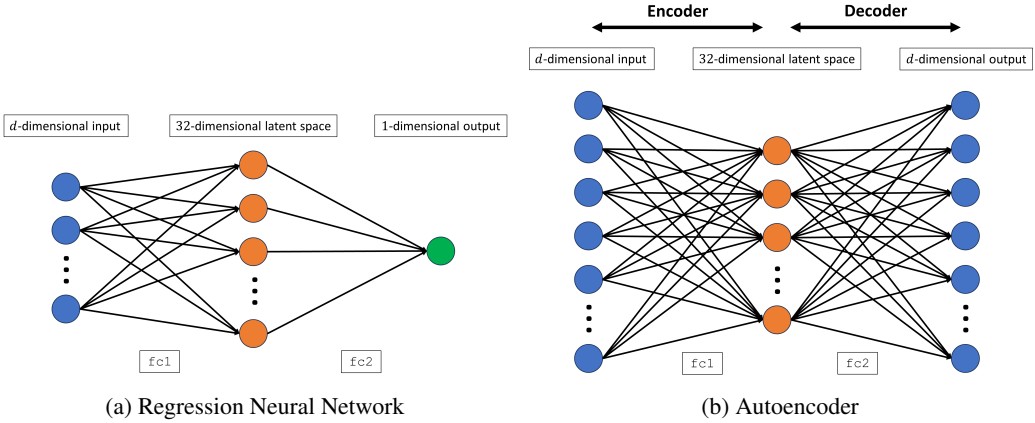

(a) Regression Neural Network        (b) Autoencoder

Figure 20: Architectures of the regression neural network (left) and autoencoder (right)

### D.1.3 LeNet-5 for Image Classification on MNIST

LeNet-5 [Lecun et al., 1998], a convolutional neural network, is widely used for image classification tasks such as handwritten digit recognition (e.g., MNIST). For our MNIST image classification study, we preliminarily zero-pad the images so they are of size $32 \times 32$. The employed architecture of LeNet-5 (see Figure 21) consists of the following layers:

- **Convolutional Layer (`conv1`)**: Applies 6 filters of size $5 \times 5$ to the input image.
- **Tanh Activation (`tanh1`)**: Applies the hyperbolic tangent activation function element-wise.
- **Average Pooling Layer (`pool1`)**: Down-samples the feature map by a factor of 2.
- **Convolutional Layer (`conv2`)**: Applies 16 filters of size $5 \times 5$.
- **Tanh Activation (`tanh`)**: Applies the hyperbolic tangent activation function element-wise.
- **Average Pooling Layer (`pool2`)**: Further down-samples the feature map by a factor of 2.

- **Flattening Layer (flatten1)**: Transforms the 2-dimensional feature map into a flat vector.
- **Fully Connected Layer (fc1)**: Transforms the flat vector to a 120-dimensional space.
- **Tanh Activation (tanh)**: Applies the hyperbolic tangent activation function element-wise.
- **Fully Connected Layer (fc2)**: Reduces the dimensionality to 84.
- **Tanh Activation (tanh)**: Applies the hyperbolic tangent activation function element-wise.
- **Fully Connected Layer (fc3)**: Produces the final classification output with 10 dimensions.

The above architecture is considered when we use cross-entropy loss for image classification on MNIST. However, when RLP loss is employed, we include a sigmoid activation layer, sig1, that follows the last fully connected layer, fc3:

- **Sigmoid Activation (sig1)**: Ensures the output values are in the range $[0, 1]$ (probabilistic classification).

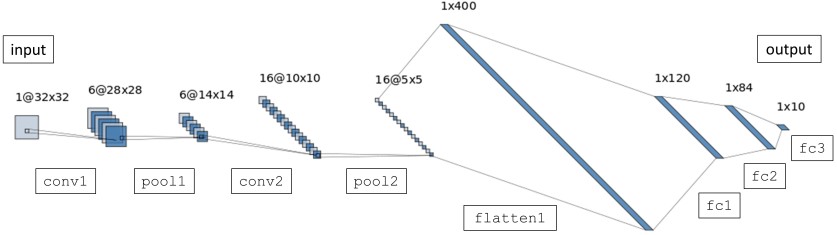

Figure 21: Architecture of LeNet-5 for image classification on MNIST

### D.1.4 MoonsClassifier for Classification on the Moons Dataset

The provided *MoonsClassifier* is a neural network designed for classifying examples from the Moons dataset, which consists of 2-dimensional points forming two interleaved half-circle shapes. The architecture of *MoonsClassifier* (see Figure 22) consists of three fully connected layers and two ReLU activation functions, as detailed below:

- **Fully Connected Layer (fc1)**: Transforms the 2-dimensional input to a 50-dimensional space. The input features represent the coordinates of a point in the dataset.
- **ReLU Activation (relu1)**: Applies the ReLU activation function element-wise, introducing non-linearity.
- **Fully Connected Layer (fc2)**: Further transforms the data in the 50-dimensional space.
- **ReLU Activation (relu2)**: Applies the ReLU activation function element-wise.
- **Fully Connected Layer (fc3)**: Reduces the dimensionality from 50 to 2, producing the final classification output.

The above architecture is considered when we use cross-entropy loss for classification on the Moons dataset. However, when RLP loss is employed, we include a sigmoid activation layer, sig1, that follows the last fully connected layer, fc3:

- **Sigmoid Activation (sig1)**: Ensures the output values are in the range $[0, 1]$ (probabilistic classification).

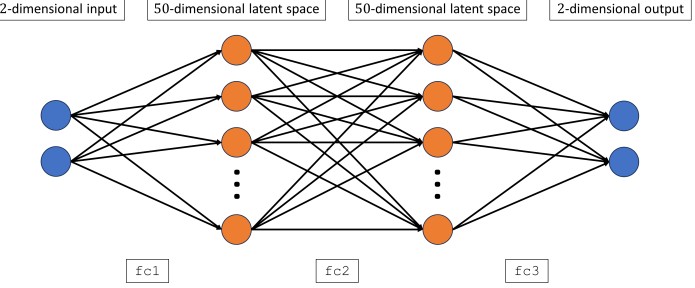

Figure 22: Architecture of *MoonsClassifier* for image classification on the Moons dataset

### D.1.5 Neural Network Training Hyperparameters

The relevant hyperparameters used to train the regression neural networks, autoencoders, and classifiers outlined in Section D.1 are provided in Tables 3 and 4. All results presented in the main text and in Sections C.1 and C.2 of the appendix were produced using these hyperparameter choices.

Table 3: Regression and reconstruction tasks — neural network training hyperparameters (grouped by dataset).

| Dataset | Experiment | Optimizer | Learning Rate ($\alpha$) | Weight Decay (MSE loss + $L_2$) | Shape Parameter ($\psi$) (Mixup & RLP + Mixup) |
|---|---|---|---|---|---|
| California Housing | No ablations | Adam | 0.0001 | 0.0001 | 0.25 |
| California Housing | $\|J\| \in \{50, 100\}$ | AdamW | 0.0005 | 0.01 | 0.25 |
| California Housing | Distribution shift | Adam | 0.0001 | 0.0001 | 0.25 |
| California Housing | Additive noise | Adam | 0.0001 | 0.0001 | 0.25 |
| California Housing | RLP vs. MAE | Adam | 0.0005 | — | — |
| California Housing | Training Time | Adam | 1.0e-6 | — | — |
| Wine Quality | No ablations | Adam | 0.0001 | 0.0001 | 0.25 |
| Wine Quality | $\|J\| \in \{50, 100\}$ | AdamW | 0.005 | 0.01 | 0.25 |
| Wine Quality | Distribution shift | Adam | 0.0001 | 0.0001 | 0.25 |
| Wine Quality | Additive noise | Adam | 0.0001 | 0.0001 | 0.25 |
| Wine Quality | RLP vs. MAE | Adam | 0.0005 | — | — |
| Wine Quality | Training time | Adam | 1.0e-6 | — | — |
| Linear | No ablations | Adam | 0.0001 | 0.0001 | — |
| Linear | $\|J\| \in \{50, 100\}$ | AdamW | 0.0005 | 0.01 | — |
| Nonlinear | No ablations | Adam | 0.0001 | 0.0001 | 0.25 |
| Nonlinear | $\|J\| \in \{50, 100\}$ | AdamW | 0.0005 | 0.01 | 0.25 |
| Nonlinear | Distribution shift | Adam | 0.0001 | 0.0001 | 0.25 |
| Nonlinear | Additive noise | Adam | 0.0001 | 0.0001 | 0.25 |
| Nonlinear | RLP vs. MAE | Adam | 0.0005 | — | — |
| Nonlinear | Training time | Adam | 1.0e-6 | — | — |
| MNIST | No ablations | SGD | 0.01 | 0.0001 | — |
| MNIST | $\|J\| \in \{50, 100\}$ | SGD | 0.01 | 0.0001 | — |
| CIFAR-10 | No ablations | SGD | 0.01 | 0.0001 | — |
| CIFAR-10 | $\|J\| \in \{50, 100\}$ | SGD | 0.01 | 0.0001 | — |

Table 4: Classification tasks — neural network training hyperparameters (grouped by dataset).

| Dataset | Experiment | Optimizer | Learning Rate ($\alpha$) | Weight Decay (MSE loss + $L_2$) | Shape Parameter ($\psi$) (Mixup & RLP + Mixup) |
|---|---|---|---|---|---|
| Moons Dataset | No ablations | Adam | 0.001 | — | 0.2 |
| Moons Dataset | $\|J\| = 25$ | Adam | 0.001 | — | 0.4 |
| MNIST | No ablations | AdamW | 0.002 | — | 0.2 |
| MNIST | $\|J\| = 100$ | AdamW | 0.002 | — | 0.2 |

---

The default AdamW weight decay is set to 0.0001 in all relevant experiments and is only changed for MSE loss + $L_2$ regularization.