# OpenReview forum: "Random Linear Projections Loss for Hyperplane-Based Optimization in Neural Networks"
_auai.org/UAI/2024/Conference — UAI 2024 poster_

### Official Review · Reviewer_b8g5 · 2024-03-20

**Q2-1 Originality-Novelty:** 3
**Q2-2 Correctness-Technical Quality:** 3
**Q2-5 Clarity Of Writing:** 3

**Q1 Summary And Contributions:**

This paper provides a new loss that improves training efficiency by leveraging geometric relationships within the data.The proposed RLP loss is based on minimizing the distance between sets of hyperplanes.

**Q2-3 Extent To Which Claims Are Supported By Evidence:**

3: Good: the main claims are supported by convincing evidence (in the form of adequate experimental evaluation, proofs, (pseudo-)code, references, assumptions).

**Q2-4 Reproducibility:**

2: Fair: key resources (e.g. proofs, code, data) are unavailable but key details (e.g. proof sketches, experimental setup) are sufficiently well-described for an expert to confidently reproduce the main results.

**Q3 Main Strengths:**

1. The method is validated across various datasets, demonstrating superior performance over traditional loss functions like MSE, particularly in terms of faster convergence and robustness against noise. Demonstrated effectiveness across multiple types of tasks (regression, reconstruction, and classification) underscores the loss function's versatility.

2. The paper provides a solid theoretical foundation, detailing the properties of RLP loss and illustrating why and how it improves over traditional loss functions.

**Q4 Main Weakness:**

1. As mentioned in the manuscript, the need for matrix inversion in RLP loss calculation could pose scalability challenges, especially for large-scale problems or high-dimensional data.

2. It would be better if the paper could explore how RLP loss interacts with or complements other training strategies and regularizations other than L2, providing insights into potential synergies.

**Q5 Detailed Comments To The Authors:**

1. Beyond computational complexity, are there additional limitations of RLP loss concerning its theoretical properties?

2. In what scenarios shall we consider traditional losses instead of RLP loss?

**Q9 Complying With Reviewing Instructions:**

Yes

---

> ### Author Rebuttal · Authors · 2024-04-06
>
> Dear Reviewer b8g5,
>
>
> Thank you for your helpful feedback. Below, we have provided responses to your comments.
>
> > As mentioned in the manuscript, the need for matrix inversion in RLP loss calculation could pose scalability challenges, especially for large-scale problems or high-dimensional data.
>
> We acknowledge the concern about the scalability of RLP loss in high-dimensional settings, and note that this forms a critical part of our ongoing research. We highlight that various data-driven approaches and computational methods have been proposed to expedite the matrix inversion process, which can potentially enable RLP loss to scale more efficiently [1].
>
> > It would be better if the paper could explore how RLP loss interacts with or complements other training strategies and regularizations other than L2, providing insights into potential synergies.
>
> Apart from L2 regularization, we provide an in-depth comparison between RLP loss and mixup-augmented training approaches [2] (e.g., mixup-augmented cross entropy loss and mixup-augmented MSE loss) in Section C of the Appendix. Our empirical results demonstrate that RLP loss achieves improved performance over these mixup-augmented training approaches. Additionally, we illustrate the benefits of utilizing mixup jointly with RLP loss, which further improves the performance of RLP loss.
>
> > Beyond computational complexity, are there additional limitations of RLP loss concerning its theoretical properties?
>
> Exploring the theoretical properties of RLP loss and examining the geometry of optimization when using RLP loss present significant challenges. Given that RLP loss involves random matrices, advancing our understanding of its limitations and/or advantages necessitates further theoretical research.
>
> > In what scenarios shall we consider traditional losses instead of RLP loss?
>
> In scenarios where the dimensionality of the observed dataset is prohibitively high, training neural networks with RLP loss can require considerable computational time. As we have highlighted above (and in the main text), the scalability of RLP loss to these high-dimensional cases is subject to ongoing research.
>
> Thank you again for your valuable feedback, which has been important in helping improve our paper.
> _____
>
> **References:**
>
> [1] Fan, K., Wei, Q., Carin, L. and Heller, K.A., 2017. An inner-loop free solution to inverse problems using deep neural networks. Advances in Neural Information Processing Systems, 30.
>
> [2] Zhang, H., Cisse, M., Dauphin, Y.N. and Lopez-Paz, D., 2017. mixup: Beyond empirical risk minimization. arXiv preprint arXiv:1710.09412.

---

### Official Review · Reviewer_bani · 2024-03-22

**Q2-1 Originality-Novelty:** 3
**Q2-2 Correctness-Technical Quality:** 3
**Q2-5 Clarity Of Writing:** 3

**Q1 Summary And Contributions:**

The paper proposes a novel loss, called Random Linear Projections (RLP) loss, where they claim its superiority over commonly used losses, MSE in regression and CE in classification. Fundamental idea comes from the least squares problem, minimizing the difference of regression matrices. Performance in various downstream tasks empirically proves the effectiveness of the problem; yet the encoder being too shallow to actually be comparative with the modern deep models.

**Q2-3 Extent To Which Claims Are Supported By Evidence:**

3: Good: the main claims are supported by convincing evidence (in the form of adequate experimental evaluation, proofs, (pseudo-)code, references, assumptions).

**Q2-4 Reproducibility:**

4: Excellent: key resources (e.g. proofs, code, data) are available and key details (e.g. proof sketches, experimental setup) are comprehensively described for competent researchers to confidently and easily reproduce the main results.

**Q3 Main Strengths:**

The motivation and ideas of RLP loss is neat. It is quite mathematically solid. The experimental results on toy datasets are convincing.

**Q4 Main Weakness:**

Even though the loss seems to be usable on small dataset settings, I am not convinced whether this can be directly applicable to the large-scale settings:
- Let $X^+ := (X^T X)^{-1}X^T$ (i.e. pseudoinverse), $Y-h(X) :=\Delta$. Then, $\mathcal{L}(h) = \mathbb{E}[ || \Delta^T (X^+)^T X ||^2 ]$. Then, (please correct me if I'm wrong) one can think of it as reweighting each MSE loss, where reweighting is determined by the pseudoinverse of the input. However, as it depends on the least squares solution, it would presumably yield a reweighting that speeds up the optimization of the linear layer (i.e. $\min_W |XW - Y|^2$). Note that the regression experiments are done in two-layer MLP. I am not sure whether this will also work well on higher resolution X and requires much bigger encoders, such as the majority of vision tasks.
- Pseudoinverse is known to be numerically unstable in certain conditions; however, there seems to be no remedy/analysis directly targeted to this potential issue.

I was also not entirely convinced about the balanced batch generation:
- I could not find the direct implication from RLP to batch generation, possibly except for avoiding same samples within a single batch. What would be the motivation of using the balanced batch generation? Why should we follow the "uniqueness principle"?
- The generator is not hugely different from the existing common settings (let's say, torch dataloader's shuffle=True).
- If the batch generation technique is effective, the authors should've also applied with and without the technique for RLP, or MSE/CE losses, so that one can also empirically observe its effectiveness.

**Q5 Detailed Comments To The Authors:**

Further comments on method improvement
- One can also replace X with the intermediate representation of X, so to avoid (i) high dimensionality of input and (2) unnormalized input (via norm layer, etc). One can even think those as the features right before the linear layer, which will be easily applicable in existing tasks.
- Given that the model is now getting bigger, and fine-tuning large models becoming the standard approach for the downstream tasks, we can expect the proposed method to also potentially have impact in practical scenarios.

Minor comments
- Intro paragraph 3 last sentence has a missing period.
- It'd be nicer if the figures were colorblind-friendly.
- "Balanced batch generation" might require a better name. Balanced sampling usually means classwise balancing in classification setting.
- The authors should clarify BCE with CE. It seems that the authors experimented with BCE, which is known to have worse performance compared to CE, and used in multilabel classification. However, MNIST is single-label.
- For the classification task, as it outputs a single real value, it won't be easy to use softmax prob's additional properties, such as classwise confidence, etc.

**Q9 Complying With Reviewing Instructions:**

Yes

---

> ### Author Rebuttal · Authors · 2024-04-06
>
> Dear Reviewer bani,
>
> Thank you for your comprehensive and constructive feedback. We will improve the terminology and the presentation of the paper as suggested. Below, we have provided responses to your comments.
>
> > One can think of RLP loss as reweighting each MSE loss, where reweighting is determined by the pseudoinverse of the input.
>
> It is accurate to say that certain properties of RLP loss can be inferred by thinking of it as a reweighting of MSE loss, as in Proposition 2.4. However, the loss itself is more complex due to the involvement of random matrices. Another perspective on RLP loss is to view it in terms of neighboring points and the hyperplanes that match them, which approximate the derivatives of a function. Thus, by minimizing RLP loss, we aim to learn a function that aligns with the derivatives observed in the data. Hence, RLP loss can also be regarded as a higher-order method compared to MSE.
>
> > Note that the regression experiments are done in two-layer MLP. I am not sure whether this will also work well on higher resolution X and requires much bigger encoders, such as the majority of vision tasks.
>
> Regarding the performance of RLP loss versus MSE loss in multi-layer settings, we present a comparison of neural networks with 3 and 5 hidden layers trained with RLP loss using 2000 batches versus MSE loss (where the dimensionality of each hidden layer is fixed), focusing on convergence time and computational costs for the California housing dataset. Performance is measured as a percentage of achieved performance at a specific epoch relative to the optimal performance (after training until convergence). We will include these results in our revised manuscript. We plan to further investigate the scalability of RLP loss for more complex tasks, such as vision and NLP applications, in future work.
>
> *Neural Network with 3 Hidden Layers*:
>
> **RLP Loss**:
> | Epochs | Performance | Computational Cost |
> |--------|-------------|--------------------|
> | 10     | 99.6%       | 25.127 s  |
> | 25     | 99.97%      | 62.235 s  |
> | 50     | 99.98%      | 122.832 s  |
> | 100    | 100%        | 247.374 s  |
>
> **MSE Loss**:
> | Epochs | Performance | Computational Cost |
> |--------|-------------|--------------------|
> | 75     | 99.15%      | 25.073 s  |
> | 190    | 99.95%      | 62.599 s   |
> | 390    | 99.975%     | 123.026 s  |
> | 790    | 100%   | 248.113 s  |
>
>
> *Neural Network with 5 Hidden Layers*:
>
> **RLP Loss**:
> | Epochs | Performance | Computational Cost |
> |--------|-------------|--------------------|
> | 10     | 97.99%      | 35.876 s  |
> | 25     | 99.77%      | 83.309 s  |
> | 50     | 99.93%      | 162.109 s   |
> | 100    | 100%        | 320.213 s  |
>
> **MSE Loss**:
> | Epochs | Performance | Computational Cost |
> |--------|-------------|--------------------|
> | 90     | 97.85%      | 33.352 s   |
> | 215    | 99.58%      | 83.277 s  |
> | 420    | 99.92%      | 162.689 s  |
> | 830    | 100%        | 319.878 s   |
>
> > Pseudoinverse is known to be numerically unstable in certain conditions; however, there seems to be no remedy/analysis directly targeted to this potential issue.
>
> Thank you for your comment; this is indeed accurate. We believe that improving the inversion algorithm and incorporating new techniques from numerical analysis to stabilize and/or accelerate matrix inversion will significantly benefit the training process involving RLP loss.
>
> > What would be the motivation of using the balanced batch generation? Why should we follow the "uniqueness principle"?
>
> Balanced batch generation, in the context of training neural networks with RLP loss, ensures that each data point is included in at least one batch, where no two batches are identical. Regarding the intuition behind balanced batch generation, we consider the following example. Suppose we have four points comprising the training dataset, $(x_1, y_1), (x_2, y_2), (x_3, y_3), (x_4, y_4)$, where $(x_1, y_1), (x_2, y_2)$ are the left two points from Figure 1 (of the main text), and $(x_3, y_3), (x_4, y_4)$ are the right two points from Figure 1. Now, consider the following two sets of hyperplanes (lines). In the first set, we consider the line passing through $(x_1, y_1), (x_2, y_2)$ and the line passing through $(x_1, y_1), (x_3, y_3)$. In the second set, we consider the line passing through $(x_1, y_1), (x_2, y_2)$ and the line passing through $(x_3, y_3), (x_4, y_4)$. We note that the former set solely characterizes the left half of the geometry of the true function, whereas the latter set characterizes both the left and right halves of the geometry of the true function. We extend this notion to large datasets via our proposed balanced batch generation scheme, wherein we ensure that every point is represented within the constructed hyperplanes. Designing mathematically founded ways of sampling batches remains a focal point of our ongoing research.
>
> (response continued in comment)

---

### Official Review · Reviewer_w9yV · 2024-03-23

**Q2-1 Originality-Novelty:** 2
**Q2-2 Correctness-Technical Quality:** 3
**Q2-5 Clarity Of Writing:** 3

**Q1 Summary And Contributions:**

This paper propose a new loss function named random linear projection (RLP) loss, which characterizes the disparity between
all conceivable predicted hyperplanes and observed hyperplanes. With proven theoretical improvement, this paper also conducts experiments to verify the gain over traditional MSE loss and BCE loss.

**Q2-3 Extent To Which Claims Are Supported By Evidence:**

3: Good: the main claims are supported by convincing evidence (in the form of adequate experimental evaluation, proofs, (pseudo-)code, references, assumptions).

**Q2-4 Reproducibility:**

3: Good: key resources (e.g. proofs, code, data) are available and key details (e.g. proofs, experimental setup) are sufficiently well-described for competent researchers to confidently reproduce the main results.

**Q3 Main Strengths:**

This paper is well-organized and easy to follow. Also, the proposed loss seems efficient and novelty.

**Q4 Main Weakness:**

The motivation seems not so clear. If I do not miss something, I think I do not fully understand why the new loss is defined in the form. Also, while the new loss appears to have more calculation, the running time is not discussed or analyzed. Besides, the improvement over existing loss schemes seems not significant in many cases.

**Q5 Detailed Comments To The Authors:**

It is not clear to me why the loss characterizes the distance of two hyperplanes. It seems that $(X^TX)^{-1}X^T(Y-h(X))$ shows the projection of $Y-h(X)$ onto the hyperplanes spanned by $X$. However, why the loss is defined as the inner product of this projected error and a random vector in $X$ isn’t thoroughly clear. Similarly, the reason why this loss is superior to others remains elusive. Proposition 2.4 only suggests that under a very specific circumstance, the RLP loss can draw the update closer to the target point compared to the MSE loss. This doesn't conclusively prove that the RLP can consistently bring about an efficient update. A more comprehensive theoretical result may provide better generalization or a superior updating trajectory for the RLP loss. In addition, as it seems that the computation of this new loss might require more time and resources, it would indeed be helpful if the paper could include some comparison data regarding the running time.

**Q9 Complying With Reviewing Instructions:**

Yes

---

> ### Author Rebuttal · Authors · 2024-04-05
>
> Dear Reviewer w9yV,
>
> Thank you for your helpful feedback. Below, we have addressed your comments.
>
> > It is not clear to me why the loss characterizes the distance of two hyperplanes. It seems that $(X^T X)^{-1} X^T (Y-h(X)) $ shows the projection of $Y-h(X)$ onto the hyperplanes spanned by X. However, why the loss is defined as the inner product of this projected error and a random vector in X isn’t thoroughly clear.
>
> The expression $(X^T X)^{-1} X^T (Y-h(X))$ projects the residuals $Y-h(X)$ onto the space spanned by $X$. In the context of simple linear regression with two observations $(m=2)$, this projection can be illustrated as: $$(\alpha_1, \alpha_2) \begin{pmatrix} x_{11} & x_{21} \\\ x_{12} & x_{22} \end{pmatrix} = (y_1, y_2).$$ By applying $(X^T X)^{-1} X^T Y$, we obtain $(\alpha_1, \alpha_2)$, thereby recovering the estimated coefficients, as well as additional coefficients through $(X^T X)^{-1} X^T h(X)$. Therefore, we can measure the distance between these coefficients, which define the hyperplanes, either directly or by sampling from the support and evaluating the distance between sampled points within that support (i.e., we multiply the coefficients with a random $X$). We choose the latter as we empirically observe that it achieves better performance.
>
> > Similarly, the reason why this loss is superior to others remains elusive. Proposition 2.4 only suggests that under a very specific circumstance, the RLP loss can draw the update closer to the target point compared to the MSE loss. This doesn't conclusively prove that the RLP can consistently bring about an efficient update. A more comprehensive theoretical result may provide better generalization or a superior updating trajectory for the RLP loss.
>
> Intuitively, by minimizing the distance between all observed and predicted hyperplanes, our approach resembles matching the convex hulls of both the predicted and observed functions for neighboring points. This characteristic renders our method higher-order compared to MSE, as we approximately minimize the distance between derivatives, which may explain its superior performance. We are in the process of formalizing this concept and plan to further explore the theoretical performance guarantees of RLP loss in future work that builds upon this paper.
>
> > In addition, as it seems that the computation of this new loss might require more time and resources, it would indeed be helpful if the paper could include some comparison data regarding the running time.
>
> While the proposed loss function is computationally more demanding than classical loss functions, neural networks trained with RLP loss observe faster convergence and greater robustness to noise and distributional shift. Below, we present a comparison of RLP loss trained with 2000 batches and MSE loss, focusing on convergence time and computational costs for the California housing dataset. Performance is measured as a percentage of achieved performance at a specific epoch relative to the optimal performance (after training until convergence). We will include these results in our revised manuscript.
>
> **RLP Loss:**
> | Epochs | Performance | Computational Cost |
> |--------|-------------|--------------------|
> | 10     | 74.19 %     | 33.357 s           |
> | 25     | 99.991 %    | 83.062 s           |
> | 50     | 99.996 %    | 165.362 s          |
> | 100    | 99.999 %    | 334.416 s          |
> | 150    | 100 %       | 504.022 s          |
>
> **MSE Loss:**
> | Epochs | Performance | Computational Cost |
> |--------|-------------|--------------------|
> | 140    | 66.59 %     | 33.352 s           |
> | 345    | 97.9 %      | 83.074 s           |
> | 690    | 99.98 %     | 165.724 s          |
> | 1380   | 99.999 %    | 334.232 s          |
> | 2130   | 100 %       | 504.112 s          |
>
> Your thorough review has greatly enhanced the quality of our manuscript, and we sincerely appreciate your detailed and valuable feedback.

---

### Official Review · Reviewer_XdxD · 2024-03-24

**Q2-1 Originality-Novelty:** 2
**Q2-2 Correctness-Technical Quality:** 1
**Q2-5 Clarity Of Writing:** 3

**Q10 Ethical Concerns:**

No.

**Q1 Summary And Contributions:**

This paper proposes a new loss function called RLP loss, tailored for capturing non-local linear properties in observed datasets. This paper provides a mathematical analysis outlining relevant properties of RLP loss, and extends this analysis via rigorous empirical testing on benchmark and synthetic datasets for regression, reconstruction, and classification tasks.

**Q2-3 Extent To Which Claims Are Supported By Evidence:**

2: Fair: the main claims are somewhat supported by evidence (but the experimental evaluation may be weak, or does not match entirely with the claims, important baselines may be missing, proofs contain important ideas but lack rigor, algorithmic details are only discussed superficially, references are imprecise, assumptions are not sufficiently motivated or explicated, etc.).

**Q2-4 Reproducibility:**

3: Good: key resources (e.g. proofs, code, data) are available and key details (e.g. proofs, experimental setup) are sufficiently well-described for competent researchers to confidently reproduce the main results.

**Q3 Main Strengths:**

1. This paper starts a new view for training NNs (although I think that some of the claims do not make sense to me).
2. The writing of this paper is clear and easy to follow.

**Q4 Main Weakness:**

1. I am not convinced by the beginning assumption in figure 1, i.e., "if two functions share the same hyperplanes connecting all subsets of their feature-label pairs, then they must necessarily be equivalent". I think that this statement is true only when we have infinite i.i.d. training samples. However, it is not feasible in practice. As shown in figure 1, the functions between training samples are not under control by the finite samples. For the toy example shown in figure 1 (i.e., 1-D square function), we may be able to hope that interpolations of the training samples can have some effects, but for modern DNNs, the functions are usually high-dimensional, nonlinear, and non-convex so that the assumption here may fail quickly.
2. I am not convinced by the non-local properties mentioned in this paper, which are just some pair-wise linear connections. However, I do not know how much these linear properties can help for the modern very deep and complex DNNs.
3. I am very concerned about the computational costs of the proposed method as the authors acknowledge. In definition 2.2, training with the proposed RLP loss needs higher-order computational costs than MSE, which is hard to be in many large-scale settings.
4. I am not convinced by the experiment settings and results. As I said above, I am very skeptical about the performance for complex nonlinear dataset and DNNs. This paper only shows some empirical results for some simple datasets, like linear datasets and MNIST, and some simple architectures in Appendix D.1.

**Q5 Detailed Comments To The Authors:**

1. See Q4.
2. This novel view may work in some cases, but I suggest that the assumptions and experiments can consider more about more realistic and complex nonlinear and non-convex cases.

**Q9 Complying With Reviewing Instructions:**

Yes

---

> ### Author Rebuttal · Authors · 2024-04-05
>
> Dear Reviewer XdxD,
>
> Thank you for your valuable feedback. Below, we have addressed your comments in order.
>
> 1. Suppose that we have two distinct functions $f: \mathbb{R} \rightarrow \mathbb{R}$ and $g: \mathbb{R} \rightarrow \mathbb{R}$. Consequently, there exists a point $x \in \mathbb{R}$ such that $f(x) \neq g(x)$. Without loss of generality, let $x\neq 0$ and consider the hyperplanes (lines in this context) defined by the points ${(x, f(x)), (0, f(0))}$ and ${(x, g(x)), (0, g(0))}$. These lines are necessarily distinct, which implies that two different functions cannot correspond to the same set of feature-label hyperplanes. Please note that this statement does not pertain to the training process or the training data; it is solely a clarification about the conditions under which two functions will be equivalent. The statement that two functions are equivalent if and only if all of their corresponding hyperplanes are equivalent is independent of the dimensionality of the two functions. This figure was solely included to help the reader understand the basic premise behind RLP loss.
>
> 2. The loss function is non-local in that it does not minimize the distance between each prediction and observation on an individual basis. Instead, it minimizes the distance between *all combinations* of observed hyperplanes and the corresponding hyperplanes of the predicted function. These hyperplanes are non-local because they are defined by a subset of data (features dimension +1), not by isolated points within the data set. Per our above response, we note that if all of the hyperplanes pertaining to the function predicted by the neural network are equivalent to all of the hyperplanes pertaining to the underlying true function, then the true and predicted functions are equivalent. This allows the neural network to learn the true function, irrespective of its nonlinearity.
>
> 3. While the proposed loss function is computationally more demanding than MSE loss, neural networks trained with RLP loss observe faster convergence and greater robustness to noise and distributional shift. Below, we present a comparison of RLP loss trained with 2000 batches and MSE loss, focusing on convergence time and computational costs for the California housing dataset. Performance is measured as a percentage of achieved performance at a specific epoch relative to the optimal performance (measured after convergence). We will include these results in our revised manuscript.
>
> **RLP Loss:**
> | Epochs | Performance | Computational Cost |
> |--------|-------------|--------------------|
> | 10     | 74.19 %     | 33.357 s           |
> | 25     | 99.991 %    | 83.062 s           |
> | 50     | 99.996 %    | 165.362 s          |
> | 100    | 99.999 %    | 334.416 s          |
> | 150    | 100 %       | 504.022 s          |
>
> **MSE Loss:**
> | Epochs | Performance | Computational Cost |
> |--------|-------------|--------------------|
> | 140    | 66.59 %     | 33.352 s           |
> | 345    | 97.9 %      | 83.074 s           |
> | 690    | 99.98 %     | 165.724 s          |
> | 1380   | 99.999 %    | 334.232 s          |
> | 2130   | 100 %       | 504.112 s          |
>
> 4. In our empirical analysis, we evaluate the performance of RLP loss on nonlinear datasets, including CIFAR10 for image reconstruction and the California Housing dataset for regression tasks. We are open to testing our method on additional datasets should the reviewers have any suggestions, and we would be happy to incorporate those results in the revised manuscript.
>
> Thank you again for your feedback, which has been instrumental in improving our paper.

---

### Official Review · Reviewer_Tpsw · 2024-03-28

**Q2-1 Originality-Novelty:** 3
**Q2-2 Correctness-Technical Quality:** 3
**Q2-5 Clarity Of Writing:** 3

**Q1 Summary And Contributions:**

This work suggests using non-local loss instead of traditional point-wise loss function to optimize neural network training. They claim that using hyperplane based MSE loss between batches allows them to approximate the non-convex loss surface faster. They also propose an algorithm to generate fixed size subsets of feature-label pairs for their loss to operate on.

**Q2-3 Extent To Which Claims Are Supported By Evidence:**

3: Good: the main claims are supported by convincing evidence (in the form of adequate experimental evaluation, proofs, (pseudo-)code, references, assumptions).

**Q2-4 Reproducibility:**

2: Fair: key resources (e.g. proofs, code, data) are unavailable but key details (e.g. proof sketches, experimental setup) are sufficiently well-described for an expert to confidently reproduce the main results.

**Q3 Main Strengths:**

The idea of using non-local hyperplane-based loss over point-wise loss, though not new, is significantly well presented in this work, compared to prior works that I have seen.
They demonstrate via extensive experimentation that the RLP loss converges faster for all datasets, and also achieves significantly lower errors (scales apart) than the compared losses.
The algorithms for distinct and limited feature-label selection, and training the neural net with the RLP loss are well explained.

**Q4 Main Weakness:**

1. Fig-1 - A neural network is acting as a function approximation. While the four hyper-planes learnt in the figure makes sense, at the training end the network would ultimately try to fit a smooth manifold over the individual manifolds learnt. However, the union of the manifold themselves does not look like a function. For a given point $x$, there seems to be a one-to-many mapping $y=f(x)$, which violates the definition of function. Am I missing the interpretation here? Please Clarify

2. The terms propositions and theorems are interchangeably used. Please stick to one keyword and justify.

3. Since MSE and MSE-L2 are point-wise losses, does measuring RLP over them signify anything? I'm not convinced that there cannot be any case where the RLP loss measured over them would turn out better.

4. The usage of the term "almost surely" in the propositions seems confusing. By the nature of the RLP loss definition, isn't it always >= 0?

5. Pg-12 Appendix - "Since the rows of $X$ are independent and identically distributed and since $M > d$, we have that $X^TX$ is full rank and invertible, and hence, $A$ is positive definite. Furthermore, $E [x_lx_j]$ are the elements of the covariance matrix of $X$, which is also positive definite." -- Yes since $M>d$, so $rank(X^TX)$ is at most $d$, but can be $<d$ as well in presence of redundant and irrelevant feature spaces. So the claim of positive-definiteness seems incorrect, should have been positive-semi-definite. Clarify

**Q5 Detailed Comments To The Authors:**

Please clarify the comments raised in the "Weakness" section

**Q9 Complying With Reviewing Instructions:**

Yes

---

> ### Author Rebuttal · Authors · 2024-04-05
>
> Dear Reviewer Tpsw,
>
> Thank you for your constructive feedback. Below, we have provided responses to your comments in order.
>
> 1. Neural networks, as depicted in Figure 1, are tasked with approximating a function $y = f(x)$, where $f: \mathbb{R}^d \to \mathbb{R}$ encapsulates the underlying relationship between $x$ and $y$. The Mean Squared Error (MSE) method minimizes the distance between observed values (labels) and model outcomes (predictions). In contrast, the Random Linear Projections (RLP) strategy aims to minimize the distance between sets of observed and predicted hyperplanes, as illustrated in Figure 1—specifically, hyperplanes determined by feature-label pairs. Therefore, the hyperplanes shown in Figure 1 should be regarded as components of the learning process, aiding in the neural network's parameter refinement. The hyperplanes do not refer to the function that the network seeks to learn, but rather serve as a means to direct the neural network towards accurately representing the manifold that best describes the observed dataset (wherein the predicted hyperplanes are equivalent to the observed hyperplanes).
>
> 2. Thank you for your remark. We will ensure consistent terminology in the revised manuscript to improve its presentation.
>
> 3. We note that both MSE loss and RLP loss are valid metrics for evaluating the performance of trained neural networks. Regarding why RLP loss is a valid metric, we note that when the distance between sets of hyperplanes pertaining to the learned function and the true function is minimized, the learned function converges to the true function. Furthermore, in order to facilitate a fair comparison between neural networks trained with different loss functions (i.e., RLP and MSE), we evaluate the performance of these networks on both metrics for completeness.
>
> 4. We note that “almost surely” in Proposition 2.3 and Proposition 2.4 does not refer to the fact that the RLP loss, $L(h) \geq 0$, and rather refers to the fact that the hypothesis minimizing the RLP loss, $h(x) = \mathbb{E}[Y | X = x]$, *holds almost surely*. It is important to note that for a subset $A \subset \mathbb{R}$ with measure zero (with respect to the probability distribution over the feature space $P$), the squared error between $h(X)$ and $Y$ over $A$ is zero, i.e., $\int_{A} (h(x) - y)^2 dP(x,y)=0 $.
>
> 5. Thank you for your comment. We will correct this in our revised manuscript.
>
> > The proposed loss empirically is performing better than compared losses. However, some of the concerns raised and lack of code (or commitment to produce it publicly upon acceptance) prevents further usage of the loss for others. The results seems very promising, but I'm sure others would run into logic reproducibility while coding, from the algorithms themselves.
>
> We appreciate the reviewer's acknowledgment of the improvements provided by our proposed loss function. We understand the concerns raised regarding the availability and reproducibility of our approach—we wish to clarify that our submission includes comprehensive and well-commented code in the supplemental materials. This code meticulously implements our proposed loss function and the associated algorithms, ensuring that readers can fully reproduce our results and methodologies. We are committed to publicly releasing this code upon acceptance of our paper.
>
> Thank you again for your comprehensive feedback, which has played a crucial role in improving our manuscript.

---

### Meta-Review · Area_Chair_zV39 · 2024-04-17

The paper proposes a new loss function, Random Linear Projections (RLP), for training neural networks. In particular, optimizing this loss minimizes the distance between the predicted hyperplanes spanned by the inputs/predictions and the associated true hyperplanes. The paper is fairly well-written, some theoretical analysis is provided along with promising experimental results, mainly on simple problems. There are concerns regarding the scalability and applicability in real scenarios, while the approach could have been motivated better. Despite these concerns, I find the main idea interesting with potential impact. However, I encourage the authors to carefully consider the comments from reviewers and the discussion to improve their paper. For example, motivating better the method, discussing in detail the current limitations, and including additional experiments.